# L-type calcium channels regulate filopodia stability and cancer cell invasion downstream of integrin signalling

Guillaume Jacquemet[1], Habib Baghirov[1,†], Maria Georgiadou[1], Harri Sihto[2], Emilia Peuhu[1], Pierre Cettour-Janet[1], Tao He[3,†], Merja Perälä[3,†], Pauliina Kronqvist[4], Heikki Joensuu[2,5] & Johanna Ivaska[1,6]

Mounting *in vitro*, *in vivo* and clinical evidence suggest an important role for filopodia in driving cancer cell invasion. Using a high-throughput microscopic-based drug screen, we identify FDA-approved calcium channel blockers (CCBs) as potent inhibitors of filopodia formation in cancer cells. Unexpectedly, we discover that L-type calcium channels are functional and frequently expressed in cancer cells suggesting a previously unappreciated role for these channels during tumorigenesis. We further demonstrate that, at filopodia, L-type calcium channels are activated by integrin inside-out signalling, integrin activation and Src. Moreover, L-type calcium channels promote filopodia stability and maturation into talin-rich adhesions through the spatially restricted regulation of calcium entry and subsequent activation of the protease calpain-1. Altogether we uncover a novel and clinically relevant signalling pathway that regulates filopodia formation in cancer cells and propose that cycles of filopodia stabilization, followed by maturation into focal adhesions, directs cancer cell migration and invasion.

[1] Turku Centre for Biotechnology, University of Turku, FIN-20520 Turku, Finland. [2] Laboratory of Molecular Oncology, Translational Cancer Biology program, University of Helsinki, FIN-00290 Helsinki, Finland. [3] VTT Medical Biotechnology, Technical Research Centre of Finland, FIN-20520 Turku, Finland. [4] Department of Pathology, University of Turku and Turku University Hospital, FIN-20520 Turku, Finland. [5] Department of Oncology, Helsinki University Hospital, FIN-00290 Helsinki, Finland. [6] Department of Biochemistry, University of Turku, FIN-20520 Turku, Finland. † Present addresses: Norwegian University of Science and Technology, NO-7491 Trondheim, Norway (H.B.); Radiometer Turku Oy, 20750 Turku, Finland (T.H.); Natural Resources Institute Finland, FI-31600 Jokioinen, Finland (M.P.). Correspondence and requests for materials should be addressed to G.J. (email: guillaume.jacquemet@utu.fi) or to J.I. (email: Johanna.ivaska@utu.fi).

Cell motility is involved at every stage of tumorigenesis and contributes to primary tumour growth, cancer cell dissemination and metastasis formation[1,2]. As metastasis remains the leading cause of cancer-related morbidity in patients with solid tumours[3], there is an immediate need to gain a more comprehensive understanding of the cellular structures and signalling pathways that drive cancer cell migration. To migrate, cells interact and sense the surrounding extracellular matrix (ECM) via transmembrane adhesion receptors such as integrins[4–6]. Integrin function is controlled by a conformational switch between active and inactive states that determine ECM ligand interaction and subsequent receptor signalling[5]. Integrin activation from within the cell (integrin inside-out signalling) is promoted by several mechanisms including the Rap1-RIAM-talin pathway and leads to integrin-ECM engagement (integrin outside-in signalling) and the recruitment and activation of a large number of proteins including the oncogenic kinases focal adhesion kinase (FAK) and Src to the integrin[4,7].

Filopodia are actin-rich finger-like protrusions that extend from the plasma membrane and have been implicated in cell migration and invasion both in vitro and in vivo[8]. Filopodia are assembled at the front of invading cancer cells[8–10] and filopodia-like structures promote cancer cell survival at metastatic sites[11,12]. Several filopodia-inducing proteins such as the molecular motor myosin-X (MYO10) or the actin-bundling protein fascin promote cancer cell invasion both in vitro and in mouse models and are associated with poor patient prognosis in multiple carcinoma types[8,13,14]. Thus, interfering with filopodia formation could be a viable strategy to inhibit cancer metastasis in vivo. MYO10 is a homodimeric molecular motor which is upregulated in breast cancer where its expression correlates with mutant p53, poor prognosis and increased metastatic potential[13,15]. Monomeric MYO10 is inactive and localizes to the cytosol or to Rab7-positive vesicles[16]. MYO10 activation, promoted by PI(3,4,5)P3, results in motor dimerization and drives filopodia formation by transporting actin regulators, cell–cell adhesion receptors and integrins to filopodia tips.

Here, we describe a novel druggable and clinically relevant pathway regulating MYO10-positive filopodia formation and stability. Unbiased high-throughput microscopy screens reveal that L-type calcium channels, through regulation of calcium entry at filopodia tips, drive filopodia stabilization. Unexpectedly, L-type calcium channels are expressed and frequently altered in many human cancers and contribute to cancer cell invasion by regulating filopodia downstream of β1 integrin and Src activation.

## Results

**L-type calcium channel blockers inhibit filopodia formation.** To identify novel regulators of filopodia formation, cancer cells expressing MYO10-GFP (to induce and visualize filopodia) were treated with a library comprising over 500 compounds for 1 h and imaged using high-throughput microscopy. The number of MYO10-positive spots was automatically quantified to determine the average number of filopodia per cell (Supplementary Fig. 1A–D, see methods for details). From this screen, several L-type calcium channel blockers (CCBs) were identified as compounds that consistently inhibit filopodia formation (Supplementary Figs 1D and 2A). In validation experiments, four structurally distinct CCBs (amlodipine besylate, felodipine, manidipine dichloride and cilnidipine) were demonstrated to significantly reduce the number of MYO10-induced filopodia in breast cancer cells as efficiently as a PI3K inhibitor (positive control to block MYO10 activity[16]), whereas a treatment with zonisamide (inhibits t-type calcium channels, voltage-gated sodium channels and carbonic anhydrase) or bumetanide (inhibits the $Na^+/K^+/2Cl^-$ cotransporter) failed to affect filopodia number (Fig. 1a). Similar results were obtained in pancreatic cancer cells following CCB treatment (Supplementary Fig. 2B). In addition, overall inhibition of calcium entry into cells by EGTA-mediated chelation of extracellular calcium dramatically reduced the number of MYO10-induced filopodia (Supplementary Fig. 2C). Together, these data indicate that calcium entry into cells via L-type calcium channels positively regulates filopodia formation in cancer cells.

**L-type calcium channel expression in cancer cell lines.** The identification of L-type calcium channels as regulators of filopodia formation in cancer cells was unexpected as their expression and activity is principally thought to be restricted to excitable cells[17,18]. L-type calcium channels are composed of multiple subunits (α1, α2δ, β and γ), of which the α1 subunit forms the core channel transporting calcium across the plasma membrane and the other subunits form regulatory components[17,19]. Four different genes (CACNA1C, CACNA1D, CACNA1F and CACNA1S) encode the α1 subunit that is targeted by CCBs. Interestingly, all four genes were found to be widely expressed at variable levels across cancer cell lines[20] (Fig. 1b) regardless of their tissue of origin (Supplementary Fig. 3A–C). Furthermore, we found that L-type calcium channels are functional in cancer cell lines MDA-MB-231 and PDAC p53[R172H] as treatment with a specific L-type calcium channel activator (BAY K8644) triggered a rapid and transient increase in intracellular calcium throughout the cell body (Fig. 1c; Supplementary Fig. 3D; Supplementary Movies 1 and 2) (3–4 fold increase at 1 min post stimulation; detected with a GFP-based calcium probe[21]) and at filopodia tips (Fig. 1d). Importantly, calcium entry mediated by the L-type calcium channel activator was inhibited in the presence of an L-type calcium channel inhibitor (Fig. 1e).

**L-type calcium channels are clinically relevant in cancer.** Analysis of public datasets using cBioPortal[22,23] revealed that L-type calcium channels are commonly altered in patient samples of different cancer types (Supplementary Fig. 4A). In particular, over 29% of patient samples in the Breast Invasive Carcinoma[24] data set displayed alterations in CACNA1C, CACNA1D, CACNA1F or CACNA1S and these alterations showed a significant association with unfavourable patient survival (Supplementary Fig. 4B–D). Interestingly, the worst survival rates were observed when alterations in CACNA1C, CACNA1D and CACNA1S were analysed together (Supplementary Table 1). Furthermore, while all four L-type calcium channel α1 subunits are expressed at low levels in both healthy breast and breast carcinoma samples (IST Online), CACNA1D is the most commonly overexpressed α1 subunit in breast carcinoma while CACNA1F is often downregulated (Supplementary Fig. 4E). CACNA1D expression was also found to be upregulated in breast cancer Oncomine data sets[25] and was the most commonly expressed L-type calcium channel α1 subunit in breast and pancreatic cancer cell lines (Supplementary Fig. 3B,C). Taken together, these data indicate that L-type calcium channels are frequently altered in breast cancer samples and that alteration in these genes may correlate with poor prognosis. The expression of the individual L-type calcium channel α1 subunit in clinical samples will require further studies using specific antibodies.

**L-type calcium channels, cancer cell migration and invasion.** Filopodia support three-dimensional (3D) cell migration and cancer invasion, particularly in cancers harbouring p53

mutations[8,13]. Importantly, similar to PI3K inhibition, CCB treatment of p53 mutant breast and pancreatic cancer cells (MDA-MB-231, P53$^{R280K}$; PDAC, P53$^{R172H}$; Su.86.86, P53$^{G245S}$) significantly impaired cancer cell invasion (Fig. 2a–c), in a concentration-dependent manner (Supplementary Fig. 5A,B). In contrast, treatment with zonisamide or bumetanide had no effect on cancer cell invasion (Fig. 2a,b). In addition, upregulation of *MYO10* expression lead to filopodia formation and was sufficient

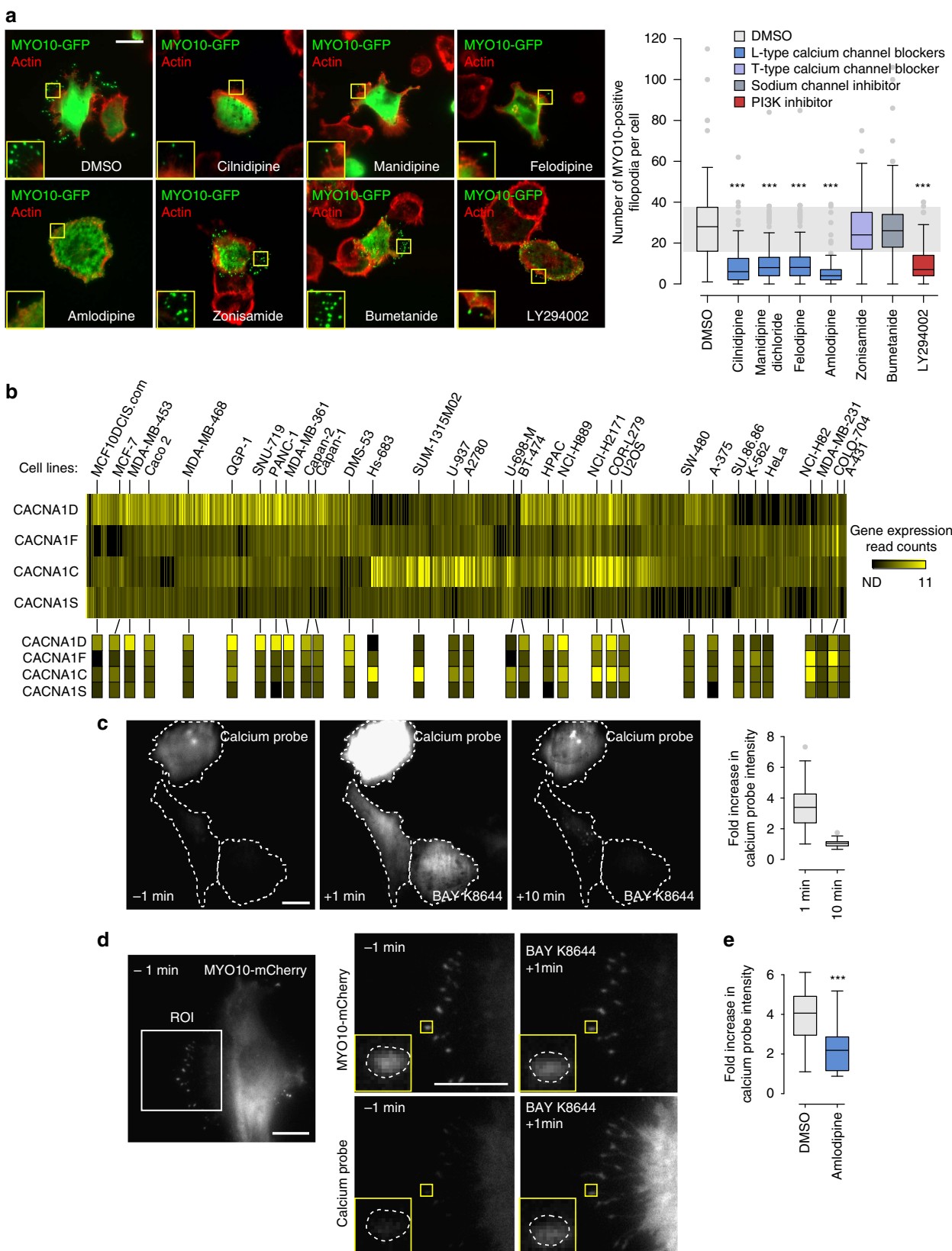

to drive cancer cell invasion in non-invasive cell lines in a CCB sensitive manner (Fig. 2d; Supplementary Fig. 5C,D). Functional L-type calcium channels and PI3K activity were also important for regulation of directionality during 3D cell migration on cell-derived matrices (Fig. 2e). Thus, *MYO10* expression facilitates filopodia formation and invasion whereas inhibition of L-type calcium channels using CCBs or inhibition of PI3K impedes filopodia formation, directional cell migration and cancer cell invasion.

Of the L-type calcium channel α1 subunits, expression of *CACNA1D*, *CACNA1S*, but not *CACNA1C*, was detected in MDA-MB-231 cells at the mRNA level (Fig. 3a; *CACNA1F* was not tested as it appeared to be often downregulated in breast cancer clinical samples). Importantly, individual silencing of *CACNA1D* or *CACNA1S* gene expression using multiple siRNA oligos (Supplementary Fig. 6A,B) decreased both filopodia formation and cancer cell invasion (Fig. 3b,c), indicating that *CACNA1D* and *CACNA1S* are likely to synergistically support filopodia formation and cancer cell invasion in MDA-MB-231 cells. The cell biological reagents to study L-type calcium channels are rather limited and thus we were only able to analyse *CACNA1D* localization (Supplementary Fig. 6B). Importantly, endogenous *CACNA1D* localized with *MYO10* at filopodia tips (Fig. 3d). In addition, *CACNA1D* protein expression could be detected in all major breast cancer subtypes (cancer type, positive expression; luminal, 3/6; triple-negative, 7/8; Her2-positive, 4/4; healthy breast, 0/1) from a set of breast carcinoma patient samples (Fig. 3e). Therefore, L-type calcium channels, in particular *CACNA1S* and *CACNA1D* are expressed in breast cancer and contribute to filopodia formation and cancer cell invasion.

**Calcium concentration and filopodia stability**. Next we sought to further study the relationship between calcium and filopodia using two distinct approaches: a GFP-based calcium probe (GCaMP6s) and a dye-based calcium indicator (Fluo4-AM). Live-cell TIRF imaging of cells transiently co-expressing the GFP-based calcium probe and *MYO10*-mCherry revealed that stable filopodia (>1 min *MYO10* spot lifetime) display higher levels of calcium than unstable filopodia (<1 min *MYO10* spot lifetime) (Fig. 4a; Supplementary Movies 3) while no difference in the fluorescence signal was observed between stable and unstable filopodia in cells expressing GFP as a control (Fig. 4b). Correspondingly, Fluo4-AM demonstrated higher intensity in stable filopodia (Supplementary Fig. 6C). These data suggest that increased calcium levels at filopodia tips correlates with filopodia stability. In line with this notion, CCB-treated cells rapidly lost most of their calcium-positive stable filopodia and displayed a higher proportion of unstable filopodia compared with DMSO-treated cells (Fig. 4c,d; Supplementary Movies 4–6). As a second read-out of filopodia stability, the average velocity of *MYO10* spots was measured and

was found to be much higher in CCB-treated cells indicative of faster filopodia turnover following L-type calcium channel inhibition (Fig. 4e). CCB treatment also decreased the number and the stability of endogenous filopodia in both breast and pancreatic cancer cell lines (see method for details; Fig. 4f,g; Supplementary Fig. 6D). Taken together, calcium entry at filopodia via L-type calcium channels contributes to filopodia stability as well as filopodia formation.

**Active integrins, filopodia and L-type calcium channels**. Integrins are one of the established cargos of *MYO10* that are transported to filopodia tips to mediate cell–ECM adhesion[26]. In addition to the filopodia tip localization we observed active β1 integrins in the shafts of *MYO10*-positive filopodia (Fig. 5a). Moreover, and congruent with a study describing Rap1-RIAM localization to filopodia[27], the integrin activator talin-1 was found to localize with *MYO10* at filopodia tips (Fig. 5b). In addition, the Rap1/talin axis was important for filopodia formation as inhibition of Rap1 (Fig. 5c; Supplementary Fig. 7A) or silencing of talin-1 expression (Fig. 5d; Supplementary Fig. 7B,C) significantly reduced filopodia number. Conversely, overexpression of a constitutively active mutant of Rap1 (CA-Rap1, Fig. 5e; Supplementary Fig. S7D) or of the FERM domain of talin-1 (talin head, Fig. 5f; Supplementary Fig. 7E), known to promote integrin activity[28], significantly increased filopodia formation. Given that Rap1 can be activated by increases in calcium levels[29], we tested whether L-type calcium channels regulate filopodia formation through Rap1. Interestingly, the expression of CA-Rap1 or talin head did not restore filopodia numbers following CCB treatment (Fig. 5e,f), suggesting that L-type calcium channels could act downstream of the Rap1-RIAM-talin pathway. In addition, CCB treatment did not inhibit overall integrin activity in cells suggesting that L-type calcium channels do not promote filopodia by inducing integrin activation (Supplementary Fig. 7F).

To further study the relationship between integrin activity, filopodia number and calcium concentration at filopodia tips, cells expressing a calcium probe and *MYO10*-mCherry were plated on conformation-specific anti-β1 integrin antibodies which lock β1 integrin in either an active or inactive conformation[30]. Using this system, β1 integrin activation significantly increased filopodia number as well as calcium levels at filopodia tips (Fig. 5g; Supplementary Fig. 8). These integrin-mediated effects were fully inhibited by CCB treatment (Fig. 5g; Supplementary Fig. 8), further demonstrating that integrin activation acts upstream of L-type calcium channels to regulate filopodia formation (Fig. 5h).

**Integrins promote filopodia formation and stability via Src**. As the activation of FAK and Src are downstream events following β1 integrin-ECM engagement, we next assessed a potential role for these kinases in filopodia formation. The Src inhibitor

**Figure 1 | L-type calcium channel blockers (CCBs) inhibit filopodia formation and L-type calcium channels are functional in cancer cells.** (**a**) MDA-MB-231 cells transiently expressing MYO10-GFP and adhering to fibronectin (FN) were treated with various compounds (10 μM) for 1 h, fixed, stained for actin and imaged on a TIRF microscope (scale bar, 20 μm). The number of MYO10-positive filopodia was counted for each cell and displayed as a box plot (three biological repeats, $n > 100$ cells, ***$P$ value $< 8.3 \times 10^{-17}$). (**b**) Relative expression of the four genes encoding the L-type calcium channel α1 subunit across 676 commonly used cancer cell lines[19]. Gene expression read counts are displayed. The value 7.99 corresponds to non-detected (ND). Selected cell lines are annotated. (**c**) MDA-MB-231 cells transiently expressing the calcium probe (pGP-CMV-GCaMP6s) and adhering to FN were treated with an L-type calcium channel activator (BAY K8644; 1 μM) while being imaged on a TIRF microscope (63× objective). The relative increase in the intracellular intensity of the calcium probe was measured at 1 and 10 min post stimulation. Cell boundaries are indicated by dotted lines (three biological repeats, $n = 74$ cells; scale bar, 20 μm). (**d**) MDA-MB-231 cells transiently expressing the calcium probe (pGP-CMV-GCaMP6s) and MYO10-mCherry were seeded on FN and treated with an L-type calcium channel activator (BAY K8644; 1 μM) while being imaged on a TIRF microscope (100× objective; scale bar, 10 μm). The inset shows a representative MYO10-positive filopodia tip delineated by a dotted line. ROI: region of interest. (**e**) MDA-MB-231 cells transiently expressing the calcium probe (pGP-CMV-GCaMP6s) and adhering to FN were treated with an L-type calcium channel activator (BAY K8644; 1 μM) in combination with DMSO or amlodipine besylate (1 μM). The relative increase in the intracellular intensity of the calcium probe was measured at 1 min (three biological repeats, $n > 21$ cells; ***$P$ value $< 1.39 \times 10^{-4}$). $P$ values were calculated using Student's $t$-test (unpaired, two-tailed, unequal variance).

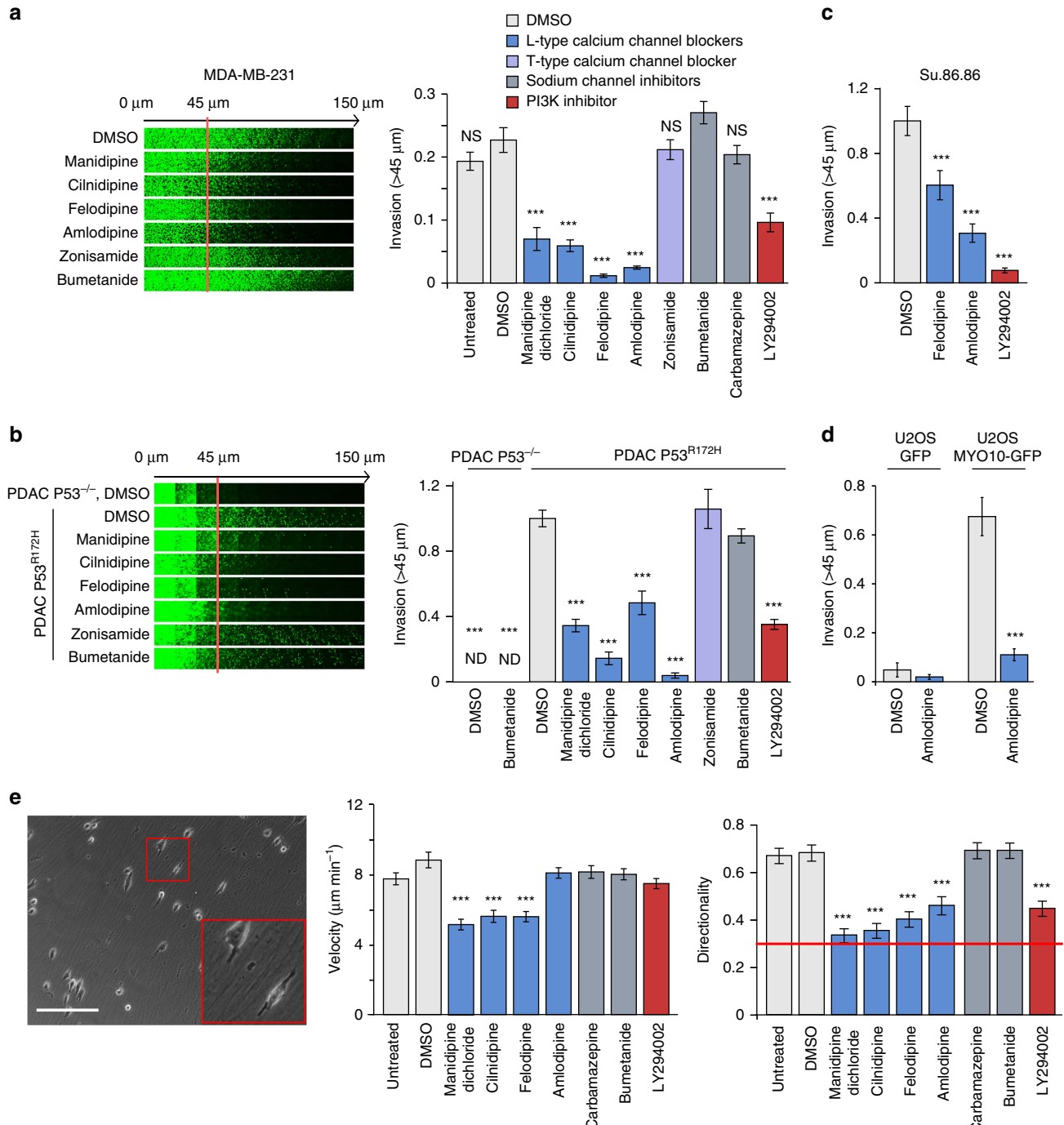

**Figure 2 | L-type calcium channels regulate filopodia formation and cancer cell invasion.** (**a**) MDA-MB-231 cells were seeded into an inverted invasion assay in the presence of various compounds (10 μM) where indicated for 48 h. The relative invasion over 45 μm was quantified ($n =$ three biological repeats, ***$P$ value $< 1.3 \times 10^{-5}$). (**b**) P53$^{-/-}$ and P53$^{R172H}$ PDAC cells were seeded into an inverted invasion assay in the presence of various compounds (10 μM) for 4 days. The relative invasion over 45 μm was quantified ($n =$ three biological repeats, ***$P$ value $< 4.1 \times 10^{-8}$). (**c**) Su.86.86 pancreatic carcinoma cells were seeded into an inverted invasion assay and allowed to invade for 4 days in the presence of various compounds (10 μM). Relative invasion over 45 μm was quantified ($n =$ three biological repeats, ***$P$ value $< 9 \times 10^{-3}$). (**d**) U2OS cells stably expressing either GFP or MYO10-GFP were seeded into an inverted invasion assay and allowed to invade for 4 days in the presence of amlodipine besylate (10 μM) or DMSO. Relative invasion over 45 μm was quantified ($n =$ three biological repeats, ***$P$ value $< 4.05 \times 10^{-6}$). (**e**) MDA-MB-231 cells were seeded on fibroblast-generated cell derived matrices (representative image is shown) in the presence of various compounds (10 μM), and cell migration was recorded over 24 h. Over 65 cells were manually tracked for each condition and migration speed and directionality were measured ($n =$ two biological repeats, scale bar, 200 μm; ***$P$ value $< 2.6 \times 10^{-5}$). $P$ values were calculated using Student's $t$-test (unpaired, two-tailed, unequal variance). All error bars represent s.e.m.

dasatinib decreased filopodia number in the original microscopy-based drug screen (60% fewer filopodia than DMSO at 20 μM). This result was validated using a small molecule inhibitor of Src (PP2) which triggered a significant loss in filopodia compared with controls (PP3 and DMSO). In contrast, inhibition of FAK had no effect on filopodia formation (Fig. 6a;

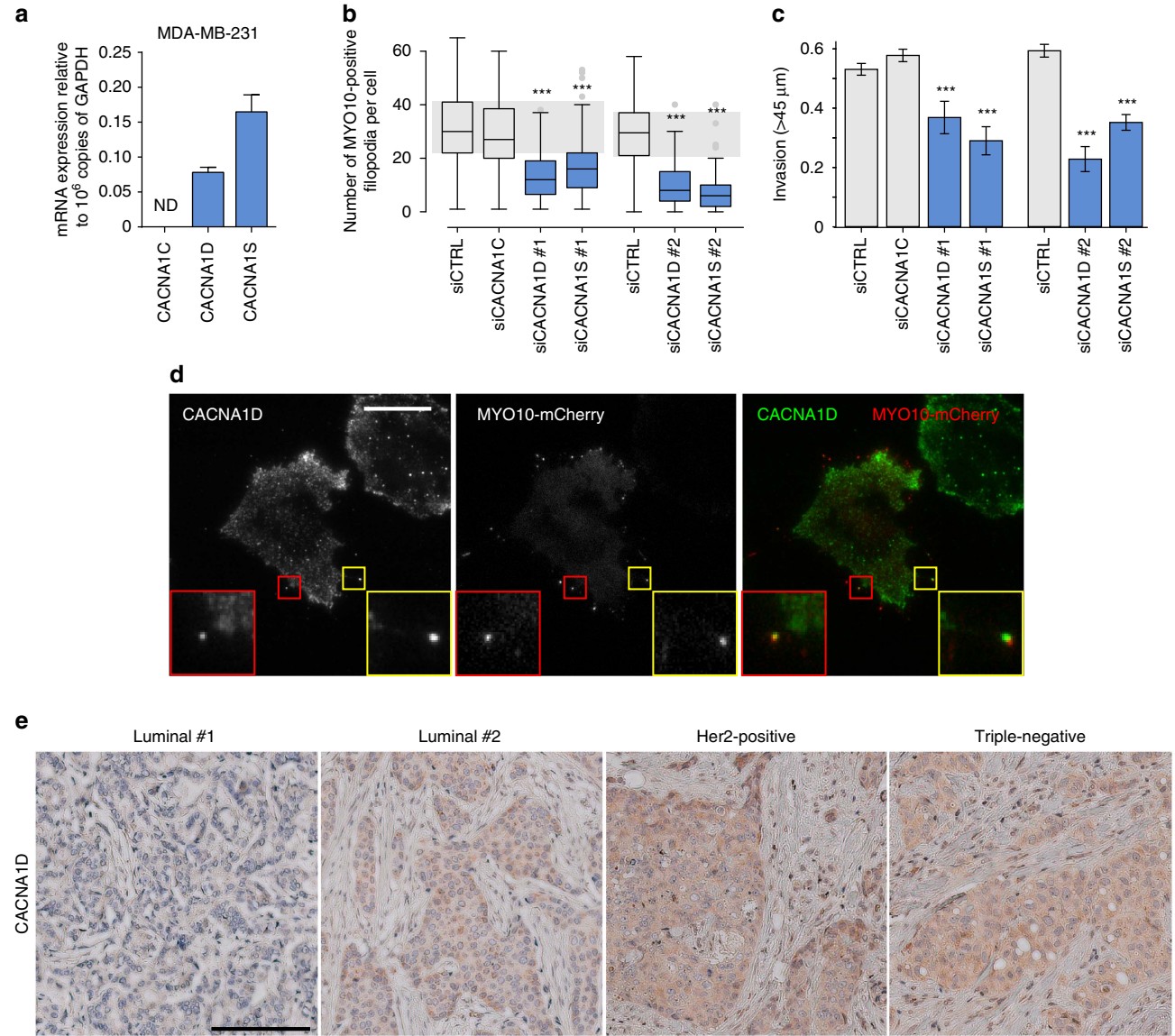

**Figure 3 | CACNA1D and CACNA1S regulate filopodia formation and cancer cell invasion.** (**a**) Relative expression of *CACNA1C*, *CACNA1D* and *CACNA1S* in MDA-MB-231 cells as determined by Q-RT-PCR ($n =$ three biological repeats). (**b**) MDA-MB-231 cells previously silenced for *CACNA1D* or *CACNA1S* using multiple RNAi oligos as indicated and transiently expressing MYO10-GFP were plated on FN for 2 h, fixed and the number of MYO10-positive filopodia per cell was quantified ($n > 95$ cells, three biological repeats ***$P$ value $< 2.4 \times 10^{-10}$). A smartpool of oligos targeting *CACNA1C* was used as an additional control as no *CACNA1C* expression was detected in these cells. (**c**) MDA-MB-231 cells previously silenced for *CACNA1D* or *CACNA1S* using multiple oligos were seeded into an inverted invasion assay and allowed to invade for 48 h. Relative invasion over 45 μm was quantified ($n =$ three biological repeats, ***$P$ value $< 8.9 \times 10^{-3}$). (**d**) MDA-MB-231 cells transiently expressing MYO10-mCherry were plated on FN for 2 h, stained for endogenous *CACNA1D* and imaged on a TIRF microscope (scale bar, 20 μm). (**e**) Representative images of various types of breast cancer tissue samples stained for *CACNA1D* (scale bar, 200 μm). $P$ values were calculated using Student's *t*-test (unpaired, two-tailed, unequal variance). All error bars represent s.e.m.

Supplementary Fig. 9A). Similar to CCB treatment, PP2-mediated inhibition of Src also promoted filopodia instability (Supplementary Fig. 9B). Importantly, overexpression of a constitutively-active mutant of Src (CA-Src) increased filopodia number while overexpression of a dominant-negative mutant of Src (DN-Src) inhibited filopodia formation (Fig. 6b; Supplementary Fig. 9C). Immunofluorescence analyses further revealed active Src (pSrc$^{Y416}$) localization to filopodia (Fig. 6c) suggesting a central role for Src in filopodia induction. These Src-dependent effects appeared to be downstream of integrin signalling as treatment with a Src inhibitor (Fig. 6d) or overexpression of DN-Src (Fig. 6e) significantly reduced filopodia number and calcium levels at filopodia tips in cells plated on the

anti-active β1 integrin antibody (Fig. 6d,e). Correspondingly, overexpression of CA-Src was sufficient to bypass the requirement for integrin activation and promoted filopodia formation and calcium increase at filopodia tips in cells plated on the anti-inactive β1 integrin antibody (Fig. 6e). Taken together, these data indicate that Src activity plays a key role in filopodia formation and stability downstream of integrin signalling. Importantly, as CA-Src-induced filopodia remained sensitive to CCB treatment (Fig. 6b) and as CCBs did not affect overall Src activity (Supplementary Fig. 9D), these results indicate that Src acts upstream of L-type calcium channels (Fig. 6f). However, how Src promotes L-type calcium channel activation remains to be determined.

Increased *MYO10* and Src protein expression and/or activity have been reported independently in breast cancer and are linked to metastasis and poor patient survival[13,15,31]. Given that Src is critical for the formation of *MYO10*-positive filopodia, we investigated a possible association between *MYO10* and Src

protein levels and/or activity in patient samples. Using cBioPortal, we determined that increased *MYO10* levels strongly correlates with increased Src mRNA levels in the Breast Invasive Carcinoma data set (Log Odds ratio: 0.826; *P* value, <0.001). Moreover, phosphoproteomic analyses indicated that patient

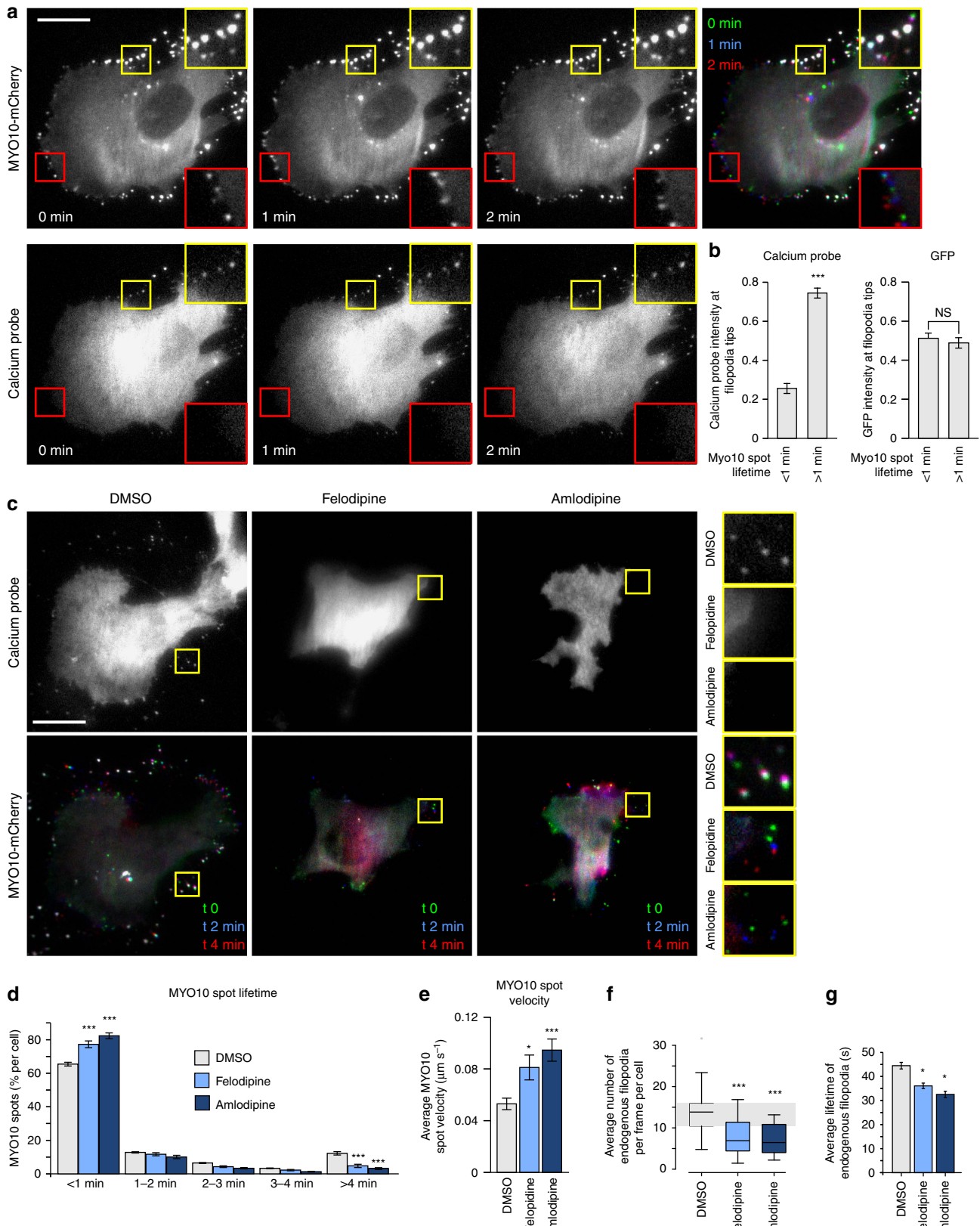

samples with high *MYO10* expression (130 patients out of 1,080; 12%) also exhibited higher Src activity (mean of alteration in altered *MYO10*, $-0.05$; mean of alteration in unaltered group, $-0.23$; *P* value, 1.408e$-$3; *q* value, 0.0162). Furthermore, re-analysis of a tissue microarray comprising 1,212 samples from a Finnish nationwide population-based breast cancer series (FinProg cohort), previously stained for *MYO10* (ref. 13) and pSrc[Y416] (refs 13,32), revealed that *MYO10* protein levels correlate with Src activity in patient samples (Fig. 6g; Table 1). Therefore, *MYO10* expression correlates with both Src expression and Src activity in breast cancer patient samples.

**Calpain 1 and filopodia formation**. Next we sought to identify the calcium-regulated pathway(s) that is/are involved in filopodia formation. Cells expressing *MYO10*-GFP were treated with inhibitors targeting multiple calcium-regulated proteins and, among these inhibitors, only the compound targeting calpain activity significantly diminished filopodia formation (Fig. 7a; Supplementary Fig. 10A). More specifically, silencing of calpain-1, and not calpain-2, expression decreased filopodia number compared with control (Fig. 7b; Supplementary Fig. 10B–E). In addition, calpain-1 and calpain-2 appeared to have cooperative functions during the invasive process as silencing of both calpain-1 and calpain-2 inhibited cell invasion more efficiently than the individual depletion of either isoform (Fig. 7c). Immunofluorescence analyses revealed that calpain-1 localizes to filopodia (Fig. 7d) and ratiometric FRET analyses of cells co-expressing a calpain sensor[33] and *MYO10*-mCherry demonstrated that calpains are active within filopodia (Fig. 7e; controls related to the use of the calpain FRET probe are shown in Supplementary Fig. 11). To assess whether calpains regulate filopodia formation downstream of integrin activation, cells expressing the calcium probe and *MYO10*-mCherry were plated on anti-β1 integrin antibodies and treated with a calpain inhibitor. Blocking calpain activity decreased filopodia number but not calcium levels at the remaining filopodia tips in cells plated on the anti-active β1 integrin antibody (Fig. 7f). Altogether, these data indicate that calpain-1 regulates filopodia formation downstream of integrin activation and calcium accumulation at filopodia tips.

**Filopodia stabilization leads to focal adhesion formation**. The importance of filopodia in the process of tumorigenesis has been described in several cancers. However, the significance of a mechanism that could regulate filopodia stability during cell migration has never been explored. CCB treatment only weakly impacted focal adhesion dynamics as their overall lifetime distribution was unaffected (Fig. 8a). CCB treatment appeared to decrease both the assembly and disassembly rate of focal

adhesions as well as their maximal size (Fig. 8a). Using live-cell TIRF imaging we found that talin-1-GFP localizes to both focal adhesions and *MYO10*-positive filopodia and in many instances was observed to move together with *MYO10* spots along filopodia shafts (Fig. 8b; Supplementary Movies 7). In stabilized filopodia, both *MYO10* and talin-1 were found to accumulate at filopodia tips (Fig. 8b; Supplementary Movies 7). Subsequent advancement of the plasma membrane coincided with *MYO10* leaving the growing talin-1-positive structure and the formation of new *MYO10*-positive filopodia followed by accumulation of a talin-positive patch resembling a classical focal adhesion. These observations are in agreement with previous studies demonstrating a role for filopodia in directing lamellipodia formation in fibroblasts[34] and maturation of filopodial shaft adhesions into focal adhesions upon lamellipodia advancement[35]. Taken together, these data allow us to propose a model by which filopodia formation and stabilization via an integrin/Src/L-type calcium channel/calpain pathway, described here, contributes to directional cell motility and cancer cell invasion (Fig. 8c).

**Discussion**

Here, we define the foundation of a druggable and clinically relevant signalling pathway (Fig. 8c) that regulates filopodia formation and stability in cancer cells. Notably, L-type calcium channels are expressed in human cancer and targeting their function with FDA-approved CCBs impairs filopodia formation and blocks cancer cell invasion. Mechanistically, we establish a link between a localized increase in calcium concentration at filopodia tips, mediated by L-type calcium channels and filopodia stability. Moreover, integrin inside-out activation and ligand binding is indispensable for filopodia formation and appears to be the first critical step in this process following *MYO10*-dependent delivery of integrins to filopodia tips. Subsequent steps require integrin outside-in signalling to trigger Src activation and the spatially restricted calcium entry at filopodia tips and the activation of the calcium-regulated protease calpain-1. Finally, we demonstrate that filopodia stabilization appears to precede focal adhesion maturation and propose that cycles of filopodia stabilization and focal adhesion maturation direct cell migration and invasion.

The central position of L-type calcium channels in cancer cell invasion was very unexpected as these voltage-gated channels (expression and activity) are considered to be restricted to excitable cells (neuronal and muscle cells). Nevertheless, several studies report roles for L-type calcium channels in many other cell types including fibroblasts, kidney cells and endometrial and prostate cancer cells[36–38]. In addition, analyses of available gene expression data sets by us (this study) and others[25] revealed a wide expression of L-type calcium channels in many cancer cell

**Figure 4 | Calcium at filopodia tips regulates filopodia stability.** (**a**) MDA-MB-231 cells transiently expressing the calcium probe (pGP-CMV-GCaMP6s) and MYO10-mCherry were plated on FN and imaged live using a TIRF microscope (1 picture every 5 s; scale bar, 20 μm). The intensity of the calcium probe at MYO10-positive filopodia tips was measured and compared between transient (<1 min lifetime) and stable filopodia (>1 min lifetime) ($n = 276$ filopodia, three biological repeats, ***P value $< 1.8 \times 10^{-9}$). (**b**) MDA-MB-231 cells transiently expressing GFP and MYO10-mCherry were plated on FN and imaged live using a TIRF microscope. The intensity of GFP at MYO10-positive filopodia tips was measured and compared between transient and stable filopodia ($n = 293$ filopodia, three biological repeats). (**c–e**) MDA-MB-231 cells transiently expressing the calcium probe (pGP-CMV-GCaMP6s) and MYO10-mCherry were plated on FN, treated with DMSO, felodipine or amlodipine besylate (10 μM) and imaged live using a TIRF microscope (1 picture every 5 s; scale bar, 20 μm). Representative images are shown (**c**). For each condition, MYO10-positive particles were automatically tracked and MYO10 spot lifetime (calculated as a percentage of the total number of filopodia generated per cell) (**d**) and average MYO10 spot velocity (**e**) were plotted (see method for details; three biological repeats, $n > 5,600$ particles tracked in more than 16 cells, *P value $= 0.015$, ***P value $< 2.98 \times 10^{-5}$). (**f,g**) MDA-MB-231 cells transiently expressing lifeact-GFP were plated on FN, treated with DMSO, felodipine or amlodipine besylate (10 μM), and imaged live on a TIRF microscope. Movies were segmented and endogenous filopodia automatically identified using CellGeo[51] (See method for details). Average filopodia number per frame and per cell (**f**) and average filopodia lifetimes are displayed (**g**) ($n > 16$ cells, two biological repeats; *P value $= 0.035$, ***P value $< 5.6 \times 10^{-5}$). *P* values were calculated using Student's *t*-test (unpaired, two-tailed, unequal variance). All error bars represent sem.

lines and clinical samples. In particular, *CACNA1D*, which localizes to filopodia tips and contributes to cancer cell invasion in vitro (shown in this study), was found to be expressed in patient samples in all major breast cancer subtypes.

The pathway delineated here as fundamental for filopodia formation in cancer cells shares some similarities with calcium-regulated processes identified in normal excitable cells. In rat smooth muscle cells α5β1 integrin signalling has been

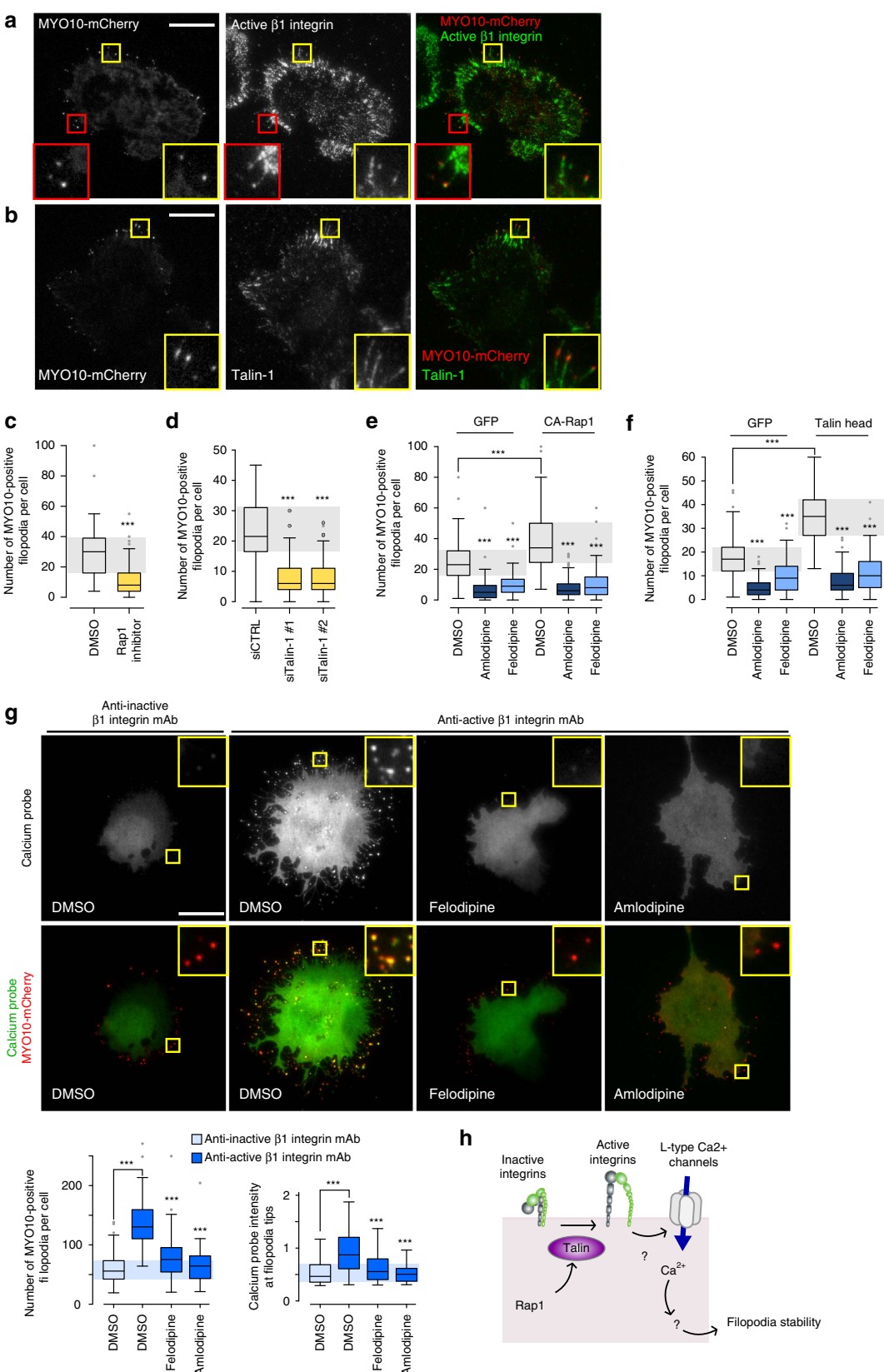

described to promote *CACNA1C*-mediated calcium entry in cells through Src phosphorylation of *CACNA1C* on Y2122 (refs 39–41). Hovever, this phosphosite does not appear to be conserved in the *CACNA1D* or *CACNA1S* isoforms identified in our study as drivers of filopodia formation in cancer cells. Whether integrins promote calcium entry through another, yet unidentifed, Src phosphorylation site within *CACNA1D* or *CACNA1S* or through another mechanism remains to be determined. In neurons, calcium signalling, and in particular L-type calcium channels and calpains, have been implicated in filopodia formation and stability in growth cones[42]. However, our discovery that non-excitable carcinoma cells have the potential to employ these calcium-dependent processes in malignant invasion was unanticipated. In particular, the relationship between calcium concentration at filopodia tips and filopodia stability in cancer cells is intriguing and appears to be important for directing the migratory process during cancer cell invasion. In neurons, the formation of dendritic filopodia is induced by small local spikes in calcium concentration and is further stabilized by accumulation of calcium in the newly formed filopodia[43,44]. The parallels between filopodia formation and stability in neurons and cancer cells could be another example of how cancer cells can hijack physiological pathways to favourably promote the metastatic process. It is noteworthy that inhibition of other calcium-regulated proteins, associated with filopodia processes in neurons, such as calcineurin and calcium-calmodulin-dependent protein kinase II (refs 44,45) does not affect filopodia formation in the MDA-MB-231 cancer cells used in this study. In the future, it will be important to define the calpain-1 target/s involved in filopodia stabilization in cancer cells.

L-type calcium channels are primarily activated by depolarizing membrane potentials, raising important questions on how these channels may be regulated in non-excitable cells. Interestingly, L-type calcium channels can be activated by rather weak depolarization events ($-55$ mV to positive)[18] and cancer cells have been reported to have depolarized membrane potential compared with their healthy counterparts[46]. In particular, MDA-MB-231 cells used in this study have a resting membrane potential of around $-20$ mV (ref. 47) which may facilitate the activation of L-type calcium channels. This would suggest that, in cancer cells with depolarized membrane potential, L-type calcium channels could be constitutively open, easily activated by small changes in membrane potential or regulated by other cues (such as activation of β1 integrin and/or Src signalling). Interestingly, a recent study identified an interaction between β1 integrin and one of the L-type calcium channel regulatory subunits in Hela cells[48], suggesting that, in addition to roles in L-type calcium channel activation, β1 integrins could also dictate L-type calcium channel sub-cellular localization.

Here, we present evidence that a pathway integral to filopodia stability is clinically relevant in breast cancer. This pathway involving integrin-driven stabilization of filopodia is potentially relevant to other carcinomas, in particular cancers harbouring p53 mutations. We have previously demonstrated that *MYO10* levels correlate with gain of function p53 mutations in human breast tumours, increased metastasis and poor prognosis. Moreover, a gain of function mutation of p53 is sufficient to promote *MYO10* expression[13]. These data are in conjunction with the observation that p53-mutated cancer cells display a high number of filopodia at the cell front during cell invasion[9,10]. Another mechanism whereby mutant p53 drives cell invasion is by promoting the co-recycling of α5β1 integrin and growth factor receptors, such as EGFR, to the cell front[6,49]. Increased EGFR signalling leads to the local activation of the PI3K/Akt pathway which, in turn, promotes filopodia formation in a RhoA- and FHOD3 formin-dependent manner[9,10]. Increased PI3K activity downstream of EGFR signalling may also support filopodia formation by promoting PIP3 production and *MYO10* activation[16]. Furthermore, increased α5β1 integrin recycling to the cell front and downstream signalling (shown in this study) will ultimately lead to an upregulation of Src activity which will enhance filopodia formation and stabilization.

Here, we propose that filopodia stability directs cell migration and promotes cancer cell invasion. The ability of filopodia to drive directional translocation of cells is probably linked to our observation that adhesions formed at filopodia tips are followed by formation of focal adhesions upon membrane advancement and subsequently give rise to new filopodia which stem from the newly formed adhesion. Given that filopodia are involved in many biological processes ranging from the development of the nervous system, to angiogenesis and cancer invasion, the delineation of a pathway regulating the individual steps of filopodia kinetics may have wide therapeutic implications. In particular, as inhibition of L-type calcium channels blocks cancer cell invasion *in vitro*, it will be important to focus further studies on the potential of these channels as cancer targets using pre-clinical cancer models.

## Methods

**Cell culture and transient transfection.** MDA-MB-231 (triple-negative human breast adenocarcinoma) cancer cells, mouse pancreatic ductal adenocarcinoma cells (PDAC p53$^{-/-}$ and PDAC p53$^{R172H}$), human telomerase immortalized foreskin fibroblasts (TIFFs) and human bone osteosarcoma epithelial cells (U2OS) were grown in DMEM (Sigma-Aldrich) supplemented with 10% FCS at 37 °C and 5% $CO_2$. The Su.86.86 human pancreatic cancer cell line was grown in DMEM (Sigma-Aldrich) supplemented with 10% FCS at 37 °C and 5% $CO_2$.

**Figure 5 | Integrin activation promotes filopodia formation and calcium entry at filopodia tips. (a,b)** MDA-MB-231 cells transiently expressing MYO10-mCherry were plated on FN, stained for active integrin (12G10 clone) (**a**) or talin-1 (**b**) and imaged using a TIRF microscope (scale bar, 20 μm). (**c**) MDA-MB-231 cells transiently expressing MYO10-GFP were plated on FN, treated with a Rap1 inhibitor (GGTI 298; 10 μM) for 1 h, fixed and the number of MYO10-positive filopodia per cell was quantified ($n > 100$ cells, three biological repeats, ***$P$ value $< 1.6 \times 10^{-14}$). (**d**) MDA-MB-231 cells previously silenced for talin-1 using two distinct siRNAs and transiently expressing MYO10-GFP were plated on FN for 2 h, fixed and the number of MYO10-positive filopodia per cell was quantified ($n > 65$ cells, three biological repeats, ***$P$ value $< 1.8 \times 10^{-24}$). (**e**) MDA-MB-231 cells transiently expressing MYO10-mCherry together with GFP or with a GFP-tagged constitutively active mutant of Rap1 (CA-Rap1) were plated on FN, treated with DMSO, felodipine or amlodipine besylate (10 μM), fixed and the number of MYO10-positive filopodia per cell was quantified ($n > 76$ cells, three biological repeats; ***$P$ value $< 4.3 \times 10^{-6}$). (**f**) MDA-MB-231 cells transiently expressing MYO10-mCherry together with GFP or with GFP-tagged talin head were plated on FN, treated with DMSO, felodipine or amlodipine besylate (10 μM), fixed and the number of MYO10-positive filopodia per cell was quantified ($n > 77$ cells, three biological repeats; ***$P$ value $< 6.29 \times 10^{-12}$). (**g**) MDA-MB-231 cells transiently expressing MYO10-mCherry and the calcium probe were plated on conformation-specific anti-β1 integrin antibodies (anti-active β1 integrin, 12G10; anti-inactive β1 integrin, 4B4) for 2 h in the presence of DMSO, felodipine or amlodipine besylate (10 μM). Representative images are displayed (scale bar, 20 μm). For each condition, the number of MYO10-positive filopodia per cell and the calcium probe intensity at filopodia tips were measured (three biological repeats, $n > 60$ cells; see Methods for details; ***$P$ value $< 1.2 \times 10^{-4}$). (**h**) Cartoon representing the sequence of events by which integrin inside-out signalling promotes calcium entry at filopodia tips, via L-type calcium channels, and subsequent filopodia stability. $P$ values were calculated using Student's $t$-test (unpaired, two-tailed, unequal variance).

The U2OS GFP and U2OS *MYO10*-GFP lines were generated by transfecting U2OS cells using lipofectamine 2000 (ThermoFisher Scientific), according to manufacturer's instructions, and selected using Geneticin (ThermoFisher Scientific; 400 μg ml$^{-1}$ final concentration). MDA-MB-231 and Su.86.86 were provided by ATCC. PDAC p53$^{-/-}$ and PDAC p53$^{R172H}$ cells were a gift from Owen Sansom (CRUK Beatson Institute, UK). TIFFs were donated by Jim Norman (CRUK Beatson Institute, UK). U2OS cells were a gift from Lea Sistonen (University of Turku, FI). All cells were tested for mycoplasma contamination.

Plasmids of interest were transfected using lipofectamine 3000 and the P3000TM Enhancer Reagent (ThermoFisher scientific) according to manufacturer's instructions. The expression of proteins of interest was suppressed using 100 nM siRNA and lipofectamine 3000 (ThermoFisher Scientific) according

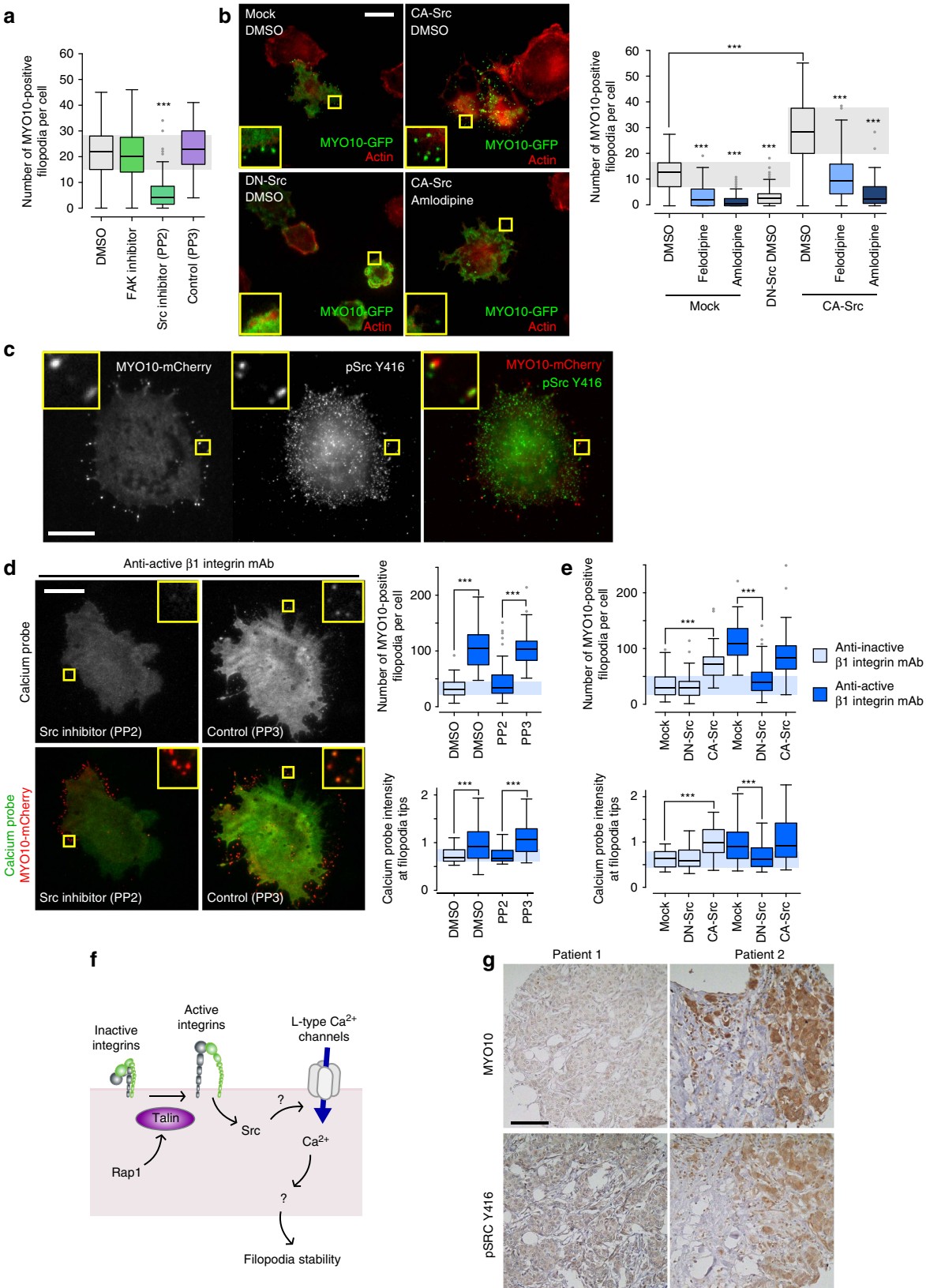

to manufacturer's instructions. SiRNA used as control (siCTRL) was Allstars negative control siRNA (Qiagen, Cat. No. 1027281). In addition, the SMARTpool ON-TARGETplus CACNA1C siRNA (Dharmacon, Cat. No. L-006123-00-0005) was used as an additional control for all experiments where SMARTpool siRNA were used. The siRNA targeting calpain-1 and calpain-2 were custom-made (Qiagen; Calpain-1 #1: 5′-AAGCTAGTGTTCGTGCACTCT-3′; Calpain-2: 5′-AAACCAGAGCTTCCAGGAAAA-3′). The siRNA targeting calpain-1 #2 was purchased from ThermoFisher Scientific (Cat. No. 4390824; siRNA ID, s490). The siRNAs targeting talin-1 were purchased from Qiagen (siTalin1#1, Hs_TLN1_2 FlexiTube siRNA, Cat. No. SI00086968; siTalin1#2, Hs_TLN1_3 FlexiTube siRNA, Cat. No. SI00086975). The siRNAs targeting CACNA1D were siCACNA1D #1 (Dharmacon, SMARTpool ON-TARGETplus, Cat. No. L-006124-00-0005) and siCACNA1D #2 (Qiagen, Hs_CACNA1D_3 FlexiTube siRNA, Cat. No. SI00337211). SiRNA targeting CACNA1S were siCACNA1S #1 (Dharmacon, SMARTpool ON-TARGETplus, Cat. No. L-006130-00-0005) and siCACNA1S #2 (Qiagen, Hs_CACNA1S_3 FlexiTube siRNA, Cat. No. SI00001260).

**Antibodies and other reagents.** Mouse anti-human β1 integrin 12G10 (ab30394; 10 μg ml$^{-1}$) and rat anti-human β1 integrin 9EG7 (553715; 1:100 for IF) monoclonal antibodies, which recognize integrin receptors with high affinity for ligand (termed 'active'), were purchased from Abcam and BD Biosciences, respectively. Mouse anti-human β1 integrin monoclonal 4B4 (10 μg ml$^{-1}$), which recognizes receptors with low affinity for ligand (termed 'inactive'), was obtained from Beckman Coulter. Mouse monoclonal antibodies raised against human talin-1 (clone 8d4, Cat. No. T3287; 1:100 for IF), human CACNA1D (used for immunocytochemistry, 1/100; clone S48A-9; Cat. No. SAB5200022), β-actin (Clone AC-15, Cat. No. A1978) and FLAG (Clone M2; Cat. No. F3165; 1:200 for IF) were purchased from Sigma. Mouse monoclonal antibodies raised against α tubulin (clone 12G10; 1:1000 for WB) and phosphotyrosine (Clone PY20; 610000; 1:1000 for WB) were purchased from the Hybridoma Bank and BD Biosciences, respectively. Rabbit polyclonal antibodies used in this study were anti-human calpain-1 (Abcam, ab39170, 1:1000 for WB), anti-human calpain-2 (Abcam, ab39165, 1:1000 for WB), anti-human CACNA1D (used for immunochemistry, 1/600; Sigma, HPA020215), anti-Phospho-Src (Y416; Cell signalling, 2101; 1:1000 for WB and 1:100 for IF), anti-human Src (Cell signalling, 2108; 1:1000 for WB) and anti-human MYO10 (Novus Biologicals, 22430002; 1:1000 for WB). Active Rap1 (CA-Rap1; pEGFP-C3-Rap1Q63E) was a gift from Buzz Baum (UCL, UK). CAAX-GFP was a gift from Gregory Giannone (Bordeaux University, France). The following plasmids were provided by Addgene (Addgene plasmid number; gift from): pCMV-calpainsensor (36182; Isabelle Richard), p3XFlag-CAPN1 (60941; Yi Zhang), pGP-CMV-GCaMP6s (40753, Douglas Kim), pLNCX chick Src E378G (13661; Joan Brugge), pCMV5 mouse Src K295R Y527F (13657; Joan Brugge and Peter Howley), mEmerald-Paxillin-22 (54219; Michael Davidson). Bovine plasma fibronectin (FN) was purchased from Merck (Cat. No. 341631). Alexa Fluor 488 and Alexa Fluor 568 phalloidin were provided by ThermoFisher Scientific. Compounds used in this study were dissolved in sterile DMSO (Sigma) and are detailed in Supplementary Table 2.

***MYO10* drug screen.** Cells stably expressing MYO10-GFP were plated at low density in 384-well plates (Corning) and treated with an FDA-approved drug library composed of 522 compounds (Selleckchem) at two different concentrations (20 and 40 μM) for 1 h. Cells were then fixed, DAPI stained and imaged using an Olympus ScanR Screening Station. Images were analysed using Fiji ImageJ packaged with Michael Schmid's Find Maxima plugin. Briefly, images were opened and, after background subtraction and normalization, MYO10 spots were automatically detected using Michael Schmid's 'Find maxima' plugin. As inactive MYO10 is known to accumulate in rab7 vesicles[16], to obtain an accurate number of filopodia-specific MYO10 spots, intracellular MYO10 spots were excluded from the analysis. Intracellular MYO10 spots were automatically filtered by analysing the

intensity of the area surrounding each spots (intracellular MYO10 spots have surrounding areas of high intensity). The remaining spots were counted and divided by the number of nuclei. The ImageJ macro used to do the quantification is provided as Supplementary Software 1.

**Immunofluorescence microscopy.** Unless otherwise stated, images were acquired on a TIRF microscope (Zeiss Laser-TIRF 3 Imaging System, Carl Zeiss) using a 63 × (numerical aperture (NA) 1.46 Oil, alpha Plan-Apochromat, DIC) or a 100 × (NA 1.46 Oil, alpha Plan-Apochromat, DIC) objective. Images were acquired on an EMCCD camera (Hamamatsu ImageEM C9100-13; Chip size 512 × 512; Hamamatsu Photonics K.K., Hamamatsu City, Japan) controlled by the Zen software (Zen 2012 Blue Edition Systems; Carl Zeiss).

For TIRF microscopy experiments, cells were plated for 2 h on glass-bottom dishes (MatTek Corporation) precoated with 10 μg ml$^{-1}$ of bovine plasma FN overnight at 4 °C. Cells were then fixed in 4% (wt/vol) paraformaldehyde (PFA) for 20 min, washed with PBS and permeabilized with PBS containing 0.5% (vol/vol) Triton X-100 for 3 min. Cells were then washed with PBS, blocked using a solution of 1 M glycine for 30 min and incubated with primary antibodies (1/100) for 30 min at room temperature, washed with PBS and incubated for a further 30 min with the appropriate secondary antibodies (1/400) at room temperature. After washing, glass-bottom dishes were stored in PBS in the dark at 4 °C before analysis.

**Filopodia formation assays on fibronectin.** Cells expressing human MYO10-GFP were plated for 2 h in full medium (1 h + 1 h treatment when stimulated with a compound of interest) on glass-bottom dishes (MatTek Corporation) precoated with 10 μg ml$^{-1}$ of FN. Cells were then fixed using PFA, washed with PBS, permeabilized and stained using phalloidin. Images were acquired on a TIRF microscope using a 63 × objective and the number of filopodia per cell was manually scored.

**Filopodia formation assays on anti-β1 integrin antibodies.** Cells expressing MYO10-mCherry and the GFP-based calcium probe (pGP-CMV-GCaMP6s) were plated for 2 h in serum-free medium on glass-bottom dishes (MatTek Corporation) precoated with 10 μg ml$^{-1}$ of conformation-specific anti-β1 integrin antibodies (active, 12G10; inactive, 4B4), overnight at 4 °C, as previously described[50]. Cells were then fixed using PFA, washed with PBS, blocked using a solution of 1 M glycine and imaged using a TIRF microscope using a 100 × objective. The quantification of the number of MYO10 spots per cell as well as the intensity of the calcium probe at filopodia tips was performed semi-automatically using the custom made ImageJ plugin provided as a supplementary file (Supplementary Software 2). This plugin requires a stack composed of two images (MYO10 and a counter-stain that allows delimitation of the cell edge). Briefly, after background subtraction, the cell edges were delimited semi-automatically using the colour threshold option. To obtain a mask of the cell that does not contain filopodia, three pixel erosion cycles followed by three pixel dilatation cycles were performed. Next, MYO10 spots were detected semi-automatically using a pass band filter followed by the identification of local maxima to create a mask containing all MYO10 spots. The cell mask and the mask containing the MYO10 spots were then compared and only the MYO10 spots located outside the cells were quantified. In addition, the integrated density of the calcium probe at MYO10 spots located outside the cell mask was measured. This plugin also calculates other parameters including the area and the min and max grey values of each spot, the minimal distance between each spot and the cell edge (filopodia length) and the number of MYO10 intracellular spots.

**Live-cell fluorescence microscopy.** For optimal image resolution, culture medium was replaced with Ham's F-12 (Gibco) containing 25 mM HEPES for all live-cell imaging experiments. For BAY K8644 stimulation experiments, cells expressing the GFP-based calcium probe (pGP-CMV-GCaMP6s) were plated for at

**Figure 6 | Src regulates filopodia formation and calcium entry at filopodia tips. (a)** MDA-MB-231 cells transiently expressing MYO10-GFP were plated on FN and treated for 1 h with DMSO, a FAK inhibitor (FAK inhibitor 14; 10 μM), a Src inhibitor (PP2; 10 μM) or a negative control for the Src inhibitor (PP3; 10 μM). The number of MYO10-positive filopodia per cell was counted (three biological repeats, $n > 66$ cells; ***P value $< 2.1 \times 10^{-25}$). **(b)** MDA-MB-231 cells transiently expressing MYO10-GFP together with PCDNA3, or with a constitutively active mutant of Src (CA-Src, Src E378G) or with a dominant negative mutant of Src (DN-Src, Src K295R Y527F) were plated on FN and treated with DMSO, felodipine or amlodipine besylate (10 μM). Representative images are displayed (scale bar, 20 μm). For each condition, the number of MYO10-positive filopodia per cell was quantified (three biological repeats, $n > 87$ cells; ***P value $< 2.5 \times 10^{-21}$). **(c)** MDA-MB-231 cells transiently expressing MYO10-mCherry were plated on FN, stained for active Src (pSrc$^{Y416}$) and imaged using a TIRF microscope (scale bar, 20 μm). **(d)** MDA-MB-231 cells transiently expressing MYO10-mCherry and the calcium probe were plated on conformation-specific anti-β1 integrin antibodies in the presence of DMSO, PP2 or PP3 (10 μM). Representative images are displayed (scale bar, 20 μm). The number of MYO10-positive filopodia per cell and the calcium probe intensity at filopodia tips were measured (three biological repeats, $n > 61$ cells; ***P value $< 2.4 \times 10^{-4}$). **(e)** MDA-MB-231 cells transiently expressing MYO10-mCherry and the calcium probe together with PCDNA3, or with CA-Src or with DN-Src, were plated on conformation-specific anti-β1 integrin antibodies. The number of MYO10-positive filopodia per cell and the calcium probe intensity at filopodia tips were measured ($n > 50$ cells, three biological repeats, ***P value $< 1.3 \times 10^{-4}$). **(f)** Cartoon displaying the sequence of events by which integrin inside-out signalling leads to filopodia stabilization. **(g)** Representative images of a breast cancer TMA that was previously stained for MYO10 and pSrc$^{Y416}$ illustrating that high MYO10 levels correlate with high Src activity in patient samples (scale bar, 100 μm). P values were calculated using Student's t-test (unpaired, two-tailed, unequal variance).

**Table 1 | MYO10 staining and pSrc $^{Y416}$ cytoplasm staining in FinProg crosstabulation.**

| MYO10 score | pSrc $^{Y416}$ cytoplasm score | | | Total |
| --- | --- | --- | --- | --- |
| | **0** | **1** | **2** | |
| 0 | | | | |
| Count | 135 | 2 | 0 | 137 |
| % with MYO10 | 98.54% | 1.45% | 0% | 100% |
| 1 | | | | |
| Count | 747 | 137 | 16 | 900 |
| % with MYO10 | 83% | 15.22% | 1.77% | 100% |
| 2 | | | | |
| Count | 123 | 46 | 6 | 175 |
| % with MYO10 | 70.28% | 26.28% | 3.42% | 100% |
| Total | | | | |
| Count | 1005 | 185 | 22 | 1212 |
| % with MYO10 | 82.9% | 15.26% | 1.81% | 100% |

least 2 h on glass-bottom dishes (MatTek Corporation) precoated with $10 \mu g \, ml^{-1}$ of FN before the start of the treatment. Cells were then imaged every 10 s, on a TIRF microscope using either a $63 \times$ or $100 \times$ objective, at $37 \, °C$ and using multi-position capabilities. BAY K8644 stimulation ($1 \mu M$ final concentration) was performed live during imaging and maximal increases in calcium were observed after 1 min stimulation. Quantification of the changes in calcium concentration, based on the calcium probe intensity, was performed using ImageJ. GCaMP6s imaging: Laser line, 488 (2.8% power); external filter, 490–550 nm; exposure time, 42 ms; EM gain, 101; large TIRF angles were used to ensure that only the cell–substrate interface was imaged. Definite focus was used to ensure that no variation occurred in the z position.

To study the relationship between filopodia stability and calcium concentration, cells were either co-expressing MYO10-mCherry and the GFP-based calcium probe (pGP-CMV-GCaMP6s) or expressing MYO10-mCherry and loaded with the calcium indicator dye Fluo-4 using the Fluo-4 Direct Calcium Assay Kit (ThermoFisher Scientific; F10471). When the GFP-based calcium probe was used cells were plated on FN for at least 2 h before the start of imaging. When Fluo-4 was used, cells were plated for 1 h on FN, before being loaded with the calcium dye according to the manufacturer's instructions (30 min at $37 \, °C + 30$ min at room temperature). Cells were imaged within 1 h from the end of Fluo-4 loading as this dye rapidly disappears from the TIRF plane and accumulates in intracellular bodies. For both the GFP-based calcium probe and the Fluo-4 approaches, cells were imaged live, every 5 s at $37 \, °C$, on a TIRF microscope using a $100 \times$ objective. GCaMP6s imaging: Laser line, 488 (5% power); external filter, 490550 nm; exposure time, 85 ms; EM gain, 363; large TIRF angles were used to ensure that only the cell-substrate interface was imaged. Single position and definite focus were used to ensure that no variation occurred in the z position. Fluo-4 imaging: Laser line, 488 (5% power); external filter, 490–550 nm; exposure time, 95 ms; EM gain, 66; large TIRF angles were used to ensure that only the cell–substrate interface was imaged. Single position and definite focus were used to ensure that no variation occurred in the z position. The calcium probe/reporter intensity at stable ($>1$ min lifetime) and unstable filopodia ($<1$ min lifetime) was measured using ImageJ. Briefly, for each cell, filopodia tips were detected, thresholded and their lifetime measured using the MYO10-mCherry channel. The integrated intensity of the calcium probe was measured at each filopodia tip. The measurements were then averaged in function of the filopodia lifetime and the ratio between calcium probe intensity at stable and unstable filopodia was calculated. The same experiment and analysis was performed in cells expressing GFP and MYO10-mCherry as a negative control using the same settings as for the GCaMP6s imaging.

To study the role of L-type calcium channels and Src in filopodia stability, cells expressing MYO10-mCherry and the GFP-based calcium probe were plated for at least 2 h on FN in the presence of the compound of interest (or DMSO) before the start of live imaging (pictures taken every 5 s at $37 \, °C$, on a TIRF microscope using a $100 \times$ objective). All MYO10 spots were then identified and tracked using the ImageJ plugin Particle Tracker 2D and 3D packaged within the Mosaic suite[51]. The velocity of MYO10 spots was then analysed using the ImageJ plugin chemotaxis tool.

To analyse the number and dynamics of endogenous filopodia, cells were transfected with lifeact-GFP and plated for at least 2 h on FN in the presence of the compound of interest (or DMSO) before being imaged live, every 10 s at $37 \, °C$ on a TIRF microscope using a $100 \times$ objective. Videos were then automatically analysed using the Matlab application CellGeo[52].

To analyse the dynamics of focal adhesions, cells were transfected with mEmerald-Paxillin and plated for at least 6 h on FN in the presence of amlodipine besylate (or DMSO) before being imaged live on a TIRF microscope, at $37 \, °C$, in presence of 5% $CO_2$. Images were acquired every minute for at least 3 h, using a $63 \times$ (NA 1.46 Oil, alpha Plan-Apochromat, DIC) objective and an internal Optovar ($1.6 \times$ magnification). Acquired videos were pre-processed using ImageJ and in particular, the contrast was adjusted, a smooth filter was applied and the frames were aligned using the 'rigid body algorithm'. Focal adhesion dynamics were then analysed by uploading the videos to the focal adhesion analysis server (FAAS; http://faas.bme.unc.edu/)[53] using the following settings: detection threshold, 2; Min Adhesion Size, 2 pixels; Min FA, Phase Length (10 min); Min FAAI Ratio, 3.

**FRET analysis of calpain activity.** MDA-MB-231 cells transiently expressing CFP (donor only), YFP (acceptor only), CFP and YFP (free donor and free acceptor) or pCMV-calpainsensor (CFP and YFP linked by a calpain cleavage site; low FRET = higher calpain activity) were plated on FN-coated glass-bottom dishes for 2 h at $37 \, °C$. Cells were then fixed in 4% (wt/vol) PFA for 10 min, washed with PBS and blocked using a solution of 1 M glycine for 30 min. Cells were then imaged using a confocal microscope (LSM780, Zeiss). The ZEN imaging software was used to generate and export the Fc images with intensities converted from the FRET index calculated for each pixel using the Youvan method. To validate the FRET approach, FRET signals (Fc) were measured in cells previously silenced for calpain 1 and 2 and in cells treated with the calcium ionophore Calcimycin (A23187; calpain activator). To measure the calpain activity in filopodia, cells were co-transfected with pCMV-calpainsensor and with MYO10-mCherry.

**SDS–PAGE and quantitative western blotting.** Protein extracts were separated under denaturing conditions by SDS–PAGE and transferred to nitrocellulose membranes. Membranes were blocked for 1 h at room temperature with blocking buffer (LI-COR Biosciences) and then incubated overnight at $4 \, °C$ with the appropriate primary antibody diluted in blocking buffer. Membranes were washed with PBS and then incubated with the appropriate fluorophore-conjugated secondary antibody diluted 1:5,000 in blocking buffer for 30 min. Membranes were washed in the dark and then scanned using an Odyssey infrared imaging system (LI-COR Biosciences). Band intensity was determined by digital densitometric analysis using Odyssey software. Uncropped blots are available in Supplementary Fig. 12.

**Cell migration on cell-derived matrices.** Cell-derived matrices were generated as previously described[54]. Briefly, TIFFs were seeded at a density of 50,000 cells per ml in a 24-well plate. When confluent, cells were cultured for a further 10 days, with medium being changed every 48 h to complete medium supplemented with $50 \, mg \, ml^{-1}$ ascorbic acid (Sigma-Aldrich) to ensure collagen cross-linking. Mature matrices were then denuded of cells using lysis buffer (PBS containing 20 mM $NH_4OH$ and 0.5 % (vol/vol) Triton X-100). Following PBS washes, matrices were incubated with $10 \, mg \, ml^{-1}$ DNase I (Roche) at $37 \, °C$ for 30 min. Matrices were then stored in PBS containing 1% (vol/vol) penicillin/streptomycin at $4 \, °C$ before use.

For cell migration analyses, MDA-MB-231 cells were seeded at a density of 5,000 cells per ml on cell-derived matrices and allowed to spread for 4 h. Cells were then filmed using an inverted widefield microscope (AxioCam MRm camera, EL Plan-Neofluar $10 \times /0.5$ NA objective (Carl Zeiss)) equipped with a heated chamber ($37 \, °C$) and $CO_2$ controller (5%). Images were collected every 10 min, and ten movies were generated for each condition. To assess cell migration, the speed and directionality of cells were measured using the Manual Tracking plug-in of ImageJ. Cell tracking was performed over a 24 h time period. Results were computed and analysed by the ImageJ plug-in Chemotaxis Tool.

**Inverted invasion assay.** Inverted invasion assays were modified from those described previously[9,55]. In brief, 200 µl of collagen I (concentration $5 \mu g \, ml^{-1}$; PureCol EZ Gel, Advanced BioMatrix) supplemented with $25 \mu g \, ml^{-1}$ FN was allowed to polymerize in inserts (8 µm ThinCert; Greiner bio-one) for 1 h at $37 \, °C$. Inserts were then inverted, and cells were seeded directly onto the opposite face of the filter. Transwell inserts were placed in serum-free medium, and medium supplemented with 10% FCS was placed on top of the matrix, providing a chemotactic serum gradient. Where appropriate, compounds were added to both the lower and upper chambers. Migrating cells were fixed 48–72 h after seeding using 4% PFA for 2 h, permeabilized in 0.5 % (vol/vol) Triton-X 100 for 30 min at room temperature, and stained overnight at $4 \, °C$ using Alexa Fluor 488 phalloidin. Plugs were then washed three times using PBS and imaged on a confocal microscope (LSM510; Zeiss) controlled using Zen2009 Systems Software (Carl Zeiss). Serial optical sections were captured every 15 µm using a $20 \times$ objective lens (NA 0.50 air, Plan-neofluar). Individual confocal images are presented in sequence with increasing penetrance from left to right. Invasion was quantified using the area calculator plugin in ImageJ, measuring the fluorescence intensity of cells invading 45 µm or more and expressing this as a percentage of the fluorescence intensity of all cells within the plug.

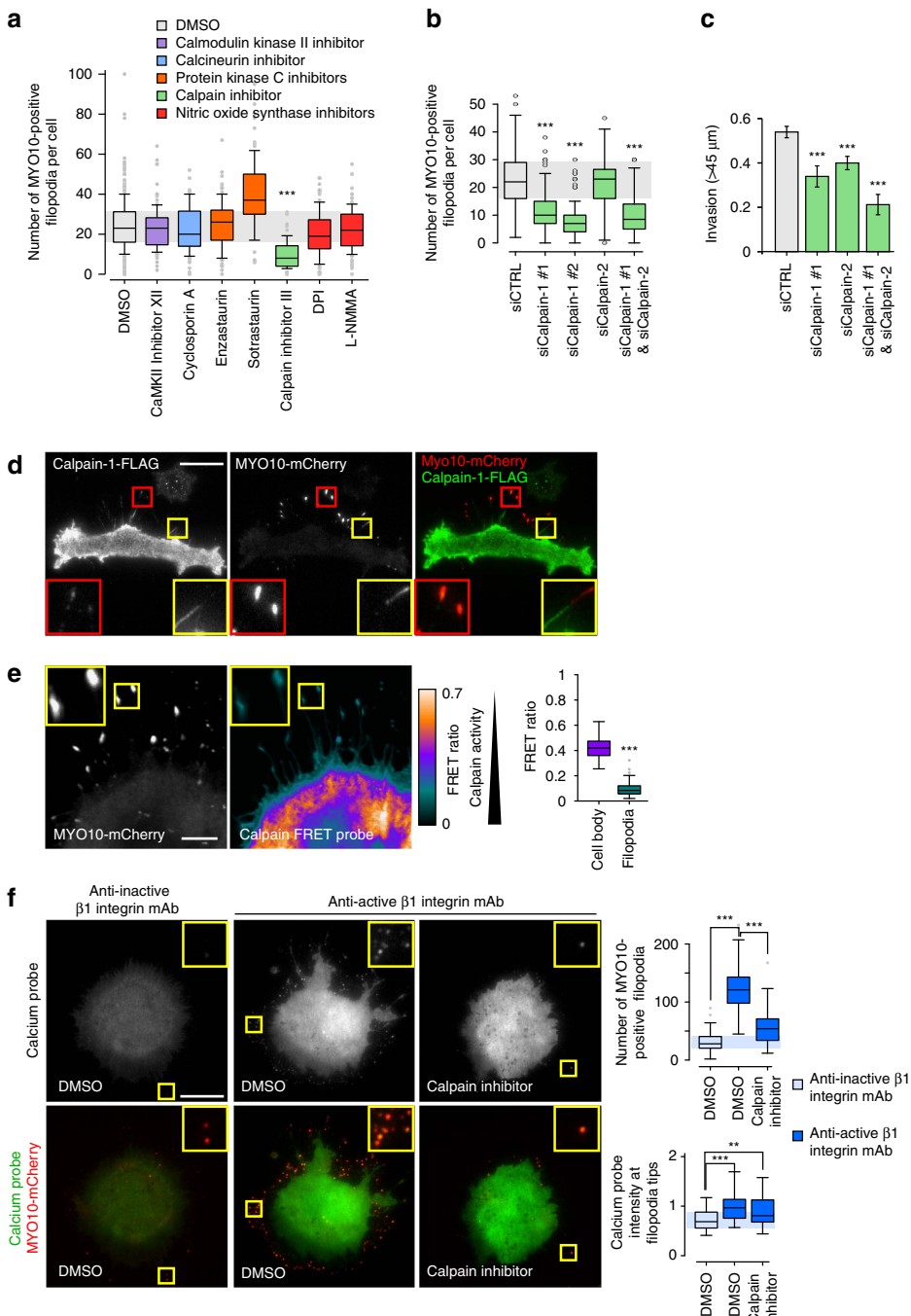

**Figure 7 | Calpain-1 regulates filopodia formation.** (**a**) MDA-MB-231 cells transiently expressing MYO10-GFP were plated on FN, treated for 1 h with various inhibitors (10 μM with the exception of cyclosporin A, enzastaurin and sotrastaurin used at 1 μM) directed against calcium-regulated pathways, fixed and imaged on a TIRF microscope. The number of MYO10-positive filopodia per cell was counted ($n > 86$ cells, three biological repeats; ***P value $< 2.4 \times 10^{-24}$). (**b**) MDA-MB-231 cells previously silenced for calpain-1, using two distinct siRNA, and/or for calpain-2, and transiently expressing MYO10-GFP, were plated on FN for 2 h, fixed and the number of MYO10-positive filopodia per cell was quantified ($n > 62$ cells, three biological repeats; ***P value $< 4.01 \times 10^{-11}$). (**c**) MDA-MB-231 cells previously silenced for calpain-1 and/or for calpain-2 were seeded into an inverted invasion assay and allowed to invade for 48 h. Relative invasion over 45 μm was quantified (three biological repeats, *** P value $< 1.2 \times 10^{-4}$). (**d**) MDA-MB-231 cells transiently expressing MYO10-mCherry and FLAG-calpain-1 were plated on FN for 2 h, stained with an anti-FLAG antibody and imaged on a TIRF microscope (scale bar, 20 μm). (**e**) MDA-MB-231 cells transiently expressing MYO10-mCherry and a calpain FRET probe (pCMV-calpainsensor; CFP and YFP linked by a calpain cleavage site, low FRET = higher calpain activity) were plated on FN and imaged on a confocal microscope (scale bar, 5 μm). The averaged FRET ratios measured in filopodia and in the cell body are displayed (three biological repeats, cell body $n = 23$, filopodia $n = 236$; ***P value $< 8.9 \times 10^{-15}$). Controls relative to this FRET experiment are presented in Supplementary Fig. 11. (**f**) MDA-MB-231 cells transiently expressing MYO10-mCherry and the calcium probe were plated on conformation-specific anti-β1 integrin antibodies (as in Fig. 5g) in the presence of DMSO or a calpain inhibitor (10 μM) (scale bar, 20 μm). The number of MYO10-positive filopodia per cell and the calcium probe intensity at filopodia tips were measured ($n > 74$ cells, three biological repeats; **P value $< 4.3 \times 10^{-3}$, ***P value $< 3.06 \times 10^{-6}$). P values were calculated using Student's t-test (unpaired, two-tailed, unequal variance).

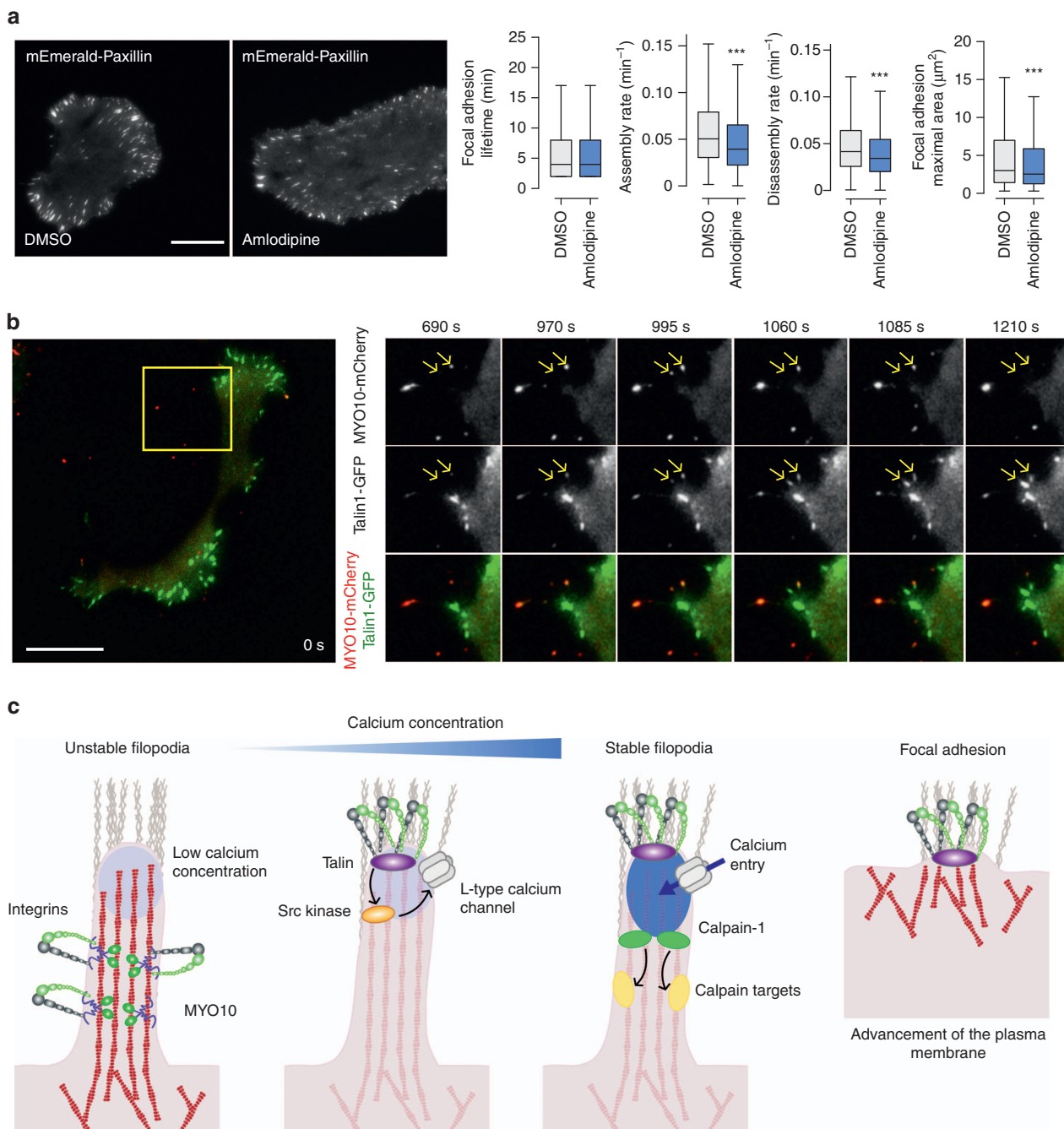

**Figure 8 | Filopodia stabilization directs cell migration. (a)** MDA-MB-231 cells transiently expressing mEmerald-Paxillin were plated on FN and imaged live using a TIRF microscope (1 picture every 1 min over 3 h; scale bar, 20 μm) in the presence of DMSO or amlodipine besylate (10 μM). Focal adhesion properties were analysed using the focal adhesion analysis server[53] (three biological repeats; over 33 movies per condition analysed; adhesion lifetime and maximal area, n > 112,000 adhesions analysed; assembly rate, n > 6100 adhesions analysed; disassembly rate, n > 7,200 adhesions analysed, ***P value < 1.73 × 10^{-50}). P values were calculated using Student's t-test (unpaired, two-tailed, unequal variance). **(b)** MDA-MB-231 cells transiently expressing talin-1-GFP and MYO10-mCherry were plated on FN and imaged live using a TIRF microscope (1 picture every 5 s; scale bar, 20 μm). Images of the region of interest (yellow squares) at the time point of interest are displayed on the right. Yellow arrows highlight MYO10-positive filopodia that precede mature focal adhesions. **(c)** Cartoon representing the sequence of events, identified in this study, leading to filopodia stabilization and ultimately focal adhesion formation. Briefly, unstable filopodia are extended from the plasma membrane to tether the surrounding environment. These filopodia display low calcium concentration at their tips. Molecular motors such as MYO10 transport integrins and other adhesion receptors to filopodia tips. Upon integrin activation, the kinase Src, directly or indirectly, promotes L-type calcium channel activity resulting in increased calcium concentration at filopodia tips. Higher calcium concentration at filopodia tips promotes calpain-1 activation and filopodia stabilization. Ultimately, upon plasma membrane advancement, adhesions initiated at filopodium tips/shafts precede mature classical focal adhesion.

**Quantitative RT-PCR.** Total RNA extracted using the NucleoSpin RNA Kit (Macherey-Nagel) was reverse transcribed into cDNA using the high-capacity cDNA reverse transcription kit (Applied Biosystems) according to the manufacturer's instructions. The RT-PCR reactions were performed using predesigned single tube TaqMan gene expression assays (GAPDH: Hs03929097_g1; CACNA1D: Hs00167753_m1; CACNA1C: Hs00167681_m1; CACNA1F:

Hs00913730_m1; CACNA1S: Hs00163885_m1) and were analysed with the 7900HT fast RT-PCR System (Applied Biosystems). Data were studied using RQ Manager Software (Applied Biosystems).

**Clinical sample analyses.** Analyses of publicly available data sets were performed using IST online (MediSapiens Ltd; http://ist.medisapiens.com/) or cBioPortal (http://www.cbioportal.org/index.do). Interrogation of the Breast Invasive Carcinoma (TCGA, Cell 2015)[24] which contains 1,105 samples was performed using cBioPortal. The genomic profiles selected were mutations, putative copy-number alterations from GISTIC and mRNA expression data (mRNA Expression z-Scores (RNA Seq V2 RSEM)). Survival analyses were performed using the Survival tab and the co-expression analyses were performed using the Co-expression or the Enrichment tabs.

Potential correlation between *MYO10* and pSrc$^{Y416}$ levels in patient samples was assessed in the FinProg series, a Finnish nationwide breast cancer cohort. Patient and sample inclusion criteria for FinProg are described in detail elsewhere[56]. The FinProg series (1,212 samples) was previously stained for pSrc$^{Y416}$ and *MYO10* expression as described earlier[13,32]. From these data a potential association between the expression groups was assessed using the $\chi^2$-test. Permission to use patient samples for research purposes was provided by the Ministry of Social Affairs and Health, Finland (permission 123/08/97).

**CACNA1D staining of patients samples.** The use of breast cancer tissue samples was performed with approval from the ethical committee of Turku University Hospital (149/6/2002) and Auria Biobank (AB15-9859) and accordingly informed consent was obtained from all human participants. Breast tissue material was prepared according to standard histology practice, that is, fixed in buffered formalin (pH 7.0) and embedded into paraffin blocks. TMAs were prepared by collecting tissue cores (4 mm in diameter) from the representative tumour area of each patient. Tissue sections (3 µm thick) were cut and stained with anti-*CACNA1D* antibody (1/600) using Lab Vision Autostainer 480 (Thermo-Fisher Scientific, Fremont, CA, USA) and detected with PowerVision+ polymer kit, according to standard protocols (DPVB+110HRP; Immunovision Technologies, Vision Biosystems, Norwell, MA, USA), using diaminobenzidine as the chromogen.

**Statistical analysis.** Statistical analyses were performed when appropriate, and *P* values indicated by an asterisk in the figure legends. Unless otherwise indicated, the Student's *t*-test was used (unpaired, two-tailed, unequal variance). Frequency tables were analysed by using two-tailed $\chi^2$-test.

**Data availability.** The authors declare that the data supporting the findings of this study are available within the article and from the authors on request.

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

## Acknowledgements

We thank J. Siivonen and P. Laasola for technical assistance, M. Saari for help with the microscopes and H. Hamidi for scientific writing and editing of the manuscript. This study has been supported by the Academy of Finland, ERC Starting Grant, ERC Consolidator Grant, the Sigrid Juselius Foundation, and the Finnish Cancer Organization. G.J. and M.G. are supported by an EMBO Long-Term Fellowship. E.P. is supported by an Academy of Finland postdoctoral fellowship.

## Author contributions

G.J designed, carried out and analysed the majority of the experiments with help from J.I, M.G, E.P and P.C. G.J wrote the manuscript with help from J.I. H.B performed the original drug screen with crucial help from T.H and M.P. H.S, H.J and P.K provided the clinical samples and the tissue sample analyses. J.I supervised the study.

## Additional information

**Competing financial interests:** The authors declare no competing financial interests.

