## [Peer Review File · Nature Communications]

Reviewers' comments:

Reviewer #1 (Remarks to the Author): Expert in ion channels and cancer

Jacquemet et al. present a study in which they report new findings on the role of voltage-gated calcium channels for the stability of filopodia and cancer cell invasion. This is a timely and well prepared study that addresses a newly emerging topic in oncology, the role of ion channels in tumor cell behavior. Based on the easy druggability of ion channels, this is particularly relevant for those channels who have a long history of being drug target, such as voltage-gated calcium channels in patients with cardiovascular diseases.

The following issues deserve clarification:

1. Pharmacology: Most of the behavioral experiments rely on a pharmacological approach that employs drug concentrations that are well beyond published IC50 values. E.g. published IC50 values of amlodipine are between 1.9 - 19.4 nM (J Cardiovasc Pharmacol. 1987 Jan;9(1):110-9). On the other hand, one of the drugs used, felodipine, was shown to block in the low micromolar range (still lower than those used in the present manuscript) a number of intracellular enzymes such as cAMP phosphodiesterase, caldesmon kinase and myosin light chain kinase that might be relevant for filopodia stabilization. Thus, there are some loose ends concerning the drug effects and the requirement for L-type calcium channels.

Bumetanide is not a "Na⁺ channel blocker" but a specific inhibitor of the Na⁺/K⁺/2Cl⁻ cotransporter. Zonisamide inhibits voltage-gated sodium channels and carbonic anhydrase, too.

2. Membrane potential measurements: Because it is not immediately intuitive that a depolarization-gated calcium channel plays a role in non-excitable cells the authors wanted to show depolarising fluctuations of the cell membrane potential. It was proposed that they could be sufficient to account for some baseline activity of the calcium channels. In the present form, the data are not very meaningful. (i) No attempt is made to calibrate the measurements. (ii) Fluctuations in filopodia and in the cell body appear to be out of phase or even inverse (fig. 1E). This cannot be given the high electrical conductance of cytosol and the small distance between the measuring points. Using the same probe, subthreshold events in neurons are propagated over much larger distances in phase (Neuron 75, 779-785, 2012). The most likely explanation is that the authors are quantifying technical noise which is larger in filopodia because of a low signal intensity as evident from CAAX-GFP probe.

Do L-type calcium channels also generate "window currents" as proposed for voltage-gated sodium channels in tumor cells?

Ref. 42 does not show that MDA-MB-231 cells used in this study have a resting membrane potential of around -20 mV - this study uses HEK293 cells.

3. Calcium measurements: More technical details for pGP-CMV-GCaMP6s calcium imaging should be given: which wavelengths, exposure time, which control for variations in the z-position. Fluo-4: "Calcium concentration at stable (>1 min lifetime) and unstable filopodia (<1min lifetime) was measured using ImageJ" -> specify how this was done; calibration etc.? ratiometric normalization to ...?

Unfortunately, no attempt is made to calibrate calcium measurements. Why? Does BayK induce an increase of Ca²⁺ in filopodia, and is it prevented by the simultaneous application of amlodipine etc.. Similarly, do the calcium channel blockers (in relevant concentrations) decrease Ca²⁺ in filopodia? Can this be mimicked by depolarising/hyperpolarising the cell membrane potential by changing the extracellular potassium concentration (in combination with potassium ionophores). Changing potassium is the classical approach to trigger L-type channel activity in smooth muscle research. Are these effects attenuated in siRNA treated cells? Omitting extracellular calcium is not specific because it also precludes calcium influx through a wide range of other calcium channels (TRPs, ORAI etc.) expressed in PDAC cells.

4. Antibody staining: How specific is the antibody? Given the notoriously poor quality of commercial antibodies against ion channels a control by Western blot etc. would have been nice.
5. Who provides calcium for RAP activation; surprising that CCB's cannot be overcome by CA-Rap. Why does integrin activation increase calcium?
6. How can the invasion phenotype be distinguished from chemotaxis defects?

Reviewer #2 (Remarks to the Author): Expert in cell motility

In this novel study, Jacquemet and colleagues use an inhibitor screen to identify a calcium-dependent pathway leading to filopodia formation and invasion of metastatic tumor cells. Further, they describe a detailed mechanism for how calcium entry through L-type calcium channels acts to stabilize focal adhesions downstream of integrin engagement. In the majority of the figures, the data is well presented and clear and supports the conclusions made by the author. In my opinion, this is an impressive body of work and it will be an important contribution to our understanding of how metastatic cells invade into the extracellular matrix. I only have a few questions and comments for the authors to address:

Questions and comments:

1. Clearly the Myo10-GFP is localizing to spots at the cell periphery. Are these spots always connected to the cell cortex via bundled F-actin as would be expected of filopodia? If so, why is this not seen in many of the images presented in the paper (eg. Figure 1A, 5B, S1E, S6A etc)?
2. Does Myo10-GFP trigger increased filopodia formation in these cells? Are these induced filopodia functionally equivalent to the endogenous filopodia in terms of facilitating L-type channel-dependent invasion? An effective way to show they are functional would be to test if Myo10-GFP expression increased cell invasion dependent on L-type calcium channel activity.
3. Pharmacological activation of the L-type calcium channels appears to increase calcium throughout the cell (Fig. 1C), but the voltage probe (Figure 1D) reports a voltage change specifically in filopodia. Can the authors account for this discrepancy and explain if the increase in calcium reported throughout the cell could be important for filopodia stabilization or invasive behavior?
4. What does time zero represent in Figure 1D? Is this prior to stimulation with the channel activator? Again, does the increased signal in the cell body also contribute to filopodia stabilization? The right panel of figure 1D (tracing the changes in intensity over time) does not appear to show any clear differences between the cell body and filopodia and weakens the impact of the figure overall. The authors might consider removing it, or finding a way to more clearly portray the differences they measured.
5. FRET is not a trivial technique. Do the authors have the appropriate positive and negative controls to be able to confirm the decrease in the calpain FRET probe signal in filpodia is due to increased calpain activity (Fig. 6E)? This is an important experiment to get right, because it provides evidence for the specificity of action of calcium in the filopodia rather than in the cell body.

Reviewer #3 (Remarks to the Author): Expert in integrins and breast cancer

This paper from the Ivaska group presents data indicating that L-type calcium channels play a key role in the formation of filopodia and the establishment of invasive behaviour in cancer cells - in particular cancer cells whose invasiveness is driven by mutations in the p53 tumour suppressor. The authors show that four genes (CACNA1C, CACNA1D, CACNA1F & CACA1S) which encode the α 1-subunit of L-type Calcium channels are expressed in cancers and cancer cell lines, and that these channels are functional in a number of cancer cells lines which have mutated p53. They then proceed to show that calcium influx through these channels occurs in filopodia (particularly in the most stable filopodia) and that blockade, siRNA or CRISPR of channel subunits destabilises filopodia, and also opposes invasiveness. From this, the authors have performed a number of mechanistic analyses that establish a novel signalling pathway linking integrin activation to filopodial stability. The data support the existence of a pathway that runs as follows: Integrin engagement.....Rap1 activation.....integrin activation.....Src activation.....opening of L-type Ca channels.....calcium influx.....calpain activation.....filopodial stability

The experiments are generally well-executed and well-controlled. The findings will be of interest to the readership of Nat. Comms., and my feeling is that that paper can be published with little extra experimentation. I do, however, have some reservations over the data that the authors use to justify a connection between expression of the CACNA1C, CACNA1D, CACNA1F & CACA1S genes and the aggressiveness of breast cancer. I think that this part of the paper needs more careful presentation of data and the statements concerning the strength of the clinical relevance of the expression of L-type calcium channels could be moderated.

I have the following comments which the authors might wish to consider:

1. The data presented in table 1 and in S3B puzzle me and I don't think that they necessarily support some of the authors' statements.

A) Firstly, is it possible to take a number of genetic aberrations that affect channel subunits - e.g. mutation, deletion, amplification - and consider them as a single group entitled 'cases with alterations in L-type calcium channels'. Maybe I'm being naïve here, but surely deletion and amplification would be expected to have opposing affects.

B) The difference between survival of patients without and with alteration to L-type Ca channels (as displayed in the KM plot in S3B and the 3rd from bottom row of table 1) appears (despite the low p value) to be very borderline. Is this really biologically significant? This is particularly in view of the fact that the median survival time of cases with unaltered query genes (column 7 of table 1) is sometimes as low as 113 months which is so close to 101 months.

C) To me the result that is more exciting is the one in the 2nd row of Table 1 showing that 'altered' CACNA1D leads to substantially reduced survival times. What would a KM curve of this look like?

D) Why has the median survival time of CACNA1F altered tumours not been determined and is displayed as NA?

D) Finally, in view of the above, I don't think that the authors can use the phrase 'Of the three clinically-relevant L-type calcium channel subunits' to justify their subsequent focus on 1D and 1S.

2. The absolute values of the number of filopodia under control conditions seem to vary quite a lot in the paper. Sometimes this is about 10 per cell (Fig. 5A), and sometimes this is much higher (Figs, 1A, 2E). I suppose that these things do vary, but this causes problems when considering the effect of integrin-activating antibodies. The authors show that, when plated onto β 1 integrin activating antibodies, cells produce about 130-140 filopodia per cell. This is fine, but the (carefully named!) anti-inactive β 1 antibody leads to at least 60 filopodia per cells, which is many more than observed in control cells. The authors perhaps need to directly compare no antibody (or irrelevant antibody), anti-inactive β 1 and anti-active β 1 directly to illustrate the point that immobilisation of integrins with either of the these antibodies tends to increase filopodia number, but to different extents.

3. Fig 1B needs some expanded panels so that expression of the channel subunits in the named cell lines can be seen.

4. The paper is left a little hanging at the end and the authors' assertion that calpain activation by calcium entry through L-type channels leads to maturation of filopodia into FAs has not been developed. Questions that spring to mind are: What is the effect of blockade of L-type channels on FA formation as cell migrate? What is the role of calpain-mediated talin cleavage in this? Some of this might be outwith the scope of the paper, but the first question should be relatively easy to address.
5. There are no scale bars on the IHC
6. What are the very stable spots in movie 5? (FEL-treated). They don't look like filopodia and their presence rather undercuts the authors' claims that FEL opposes FP stability.
7. Fig 5C doesn't look particularly convincing as displayed. It's not really possible to see the Myo10 and active colocalisation.
8. '60% less filopodia' should read '60% fewer filopodia'

Reviewer #1 (Remarks to the Author): Expert in ion channels and cancer

Jacquemet et al. present a study in which they report new findings on the role of voltage-gated calcium channels for the stability of filopodia and cancer cell invasion. This is a timely and well prepared study that addresses a newly emerging topic in oncology, the role of ion channels in tumor cell behavior. Based on the easy druggability of ion channels, this is particularly relevant for those channels who have a long history of being drug target, such as voltage-gated calcium channels in patients with cardiovascular diseases.

We thank the reviewer for his / her positive comments.

The following issues deserve clarification:

1. Pharmacology: Most of the behavioral experiments rely on a pharmacological approach that employs drug concentrations that are well beyond published IC50 values. E.g. published IC50 values of amlodipine are between 1.9 - 19.4 nM (J Cardiovasc Pharmacol. 1987 Jan;9(1):110-9). On the other hand, one of the drugs used, felodipine, was shown to block in the low micromolar range (still lower than those used in the present manuscript) a number of intracellular enzymes such as cAMP phosphodiesterase, caldesmon kinase and myosin light chain kinase that might be relevant for filopodia stabilization. Thus, there are some loose ends concerning the drug effects and the requirement for L-type calcium channels.

Bumetanide is not a "Na⁺ channel blocker" but a specific inhibitor of the Na⁺/K⁺/2Cl⁻ cotransporter. Zonisamide inhibits voltage-gated sodium channels and carbonic anhydrase, too.

We share the reviewer's concern regarding the off-target effect of inhibitors. This is why most of the experiments were carried out using four structurally distinct L-type calcium channel inhibitors and validated using two different siRNAs. Under these circumstances, it is very unlikely that the results obtained are due to off-target effects. To further rule out nonspecific effects, we have now tested several lower concentrations of amlodipine besylate on the ability to reduce filopodia formation. Our new results indicate that amlodipine besylate decreases filopodia numbers at both 1 μ M and 0.1 μ M. These data have now been included in Supplementary Figure 2A (and below for your convenience).

*Supplementary figure 2A: MDA-MB-231 cells transiently expressing MYO10-GFP and adhering to FN were treated with decreasing concentrations of Amlodipine (1 h), fixed, stained for actin and imaged on a TIRF microscope. The number of MYO10-positive filopodia was counted for each cell and displayed as a box plot ($n > 150$ cells, three biological repeats; *** p value $< 4.22 \times 10^{-22}$)*

2. Membrane potential measurements: Because it is not immediately intuitive that a depolarization-gated calcium channel plays a role in non-excitable cells the authors wanted to show depolarising fluctuations of the cell membrane potential. It was proposed that they could be sufficient to account for some baseline activity of the calcium channels. In the present form, the data are not very meaningful.

(i) No attempt is made to calibrate the measurements. (ii) Fluctuations in filopodia and in the cell body appear to be out of phase or even inverse (fig. 1E). This cannot be given the high electrical conductance of cytosol and the small distance between the measuring points. Using the same probe, subthreshold events in neurons are propagated over much larger distances in phase (Neuron 75, 779-785, 2012). The most likely explanation is that the authors are quantifying technical noise which is larger in filopodia because of a low signal intensity as evident from CAAX-GFP probe.

We agree with the reviewer and this point is well taken. This data aimed only to indicate that variations in membrane potential were observed at filopodia tips. We thought that this observation was interesting and might be linked to localised activation of the calcium channels at filopodia tips. However, as these data are not essential for the overall conclusion of the manuscript and as the same point was raised by reviewer 2 we have decided to remove this figure from the revised manuscript.

Do L-type calcium channels also generate "window currents" as proposed for voltage-gated sodium channels in tumor cells?

Window currents have also been described for L-type calcium channels. i.e. Ming et al 1994 J Cardiovasc Electrophysiol. Apr;5(4):323-34.

Formatted: Justified

"Role of L-type calcium channel window current in generating current-induced early afterdepolarizations."

Ref. 42 does not show that MDA-MB-231 cells used in this study have a resting membrane potential of around -20 mV - this study uses HEK293 cells.

We thank the reviewer for noticing this mistake. The right reference has now been cited.

Fraser S. P., Diss J. K., Chioni A. M., Mycielska M. E., Pan H., Yamaci R. F., et al. (2005). Voltage-gated sodium channel expression and potentiation of human breast cancer metastasis. Clin. Cancer Res. 11, 5381–5389 10.1158/1078-0432.CCR-05-0327 (PMID: 16061851)

3. Calcium measurements: More technical details for pGP-CMV-GCaMP6s calcium imaging should be given: which wavelengths, exposure time, which control for variations in the z-position. Fluo-4: "Calcium concentration at stable (>1 min lifetime) and unstable filopodia (<1min lifetime) was measured using ImageJ" -> specify how this was done; calibration etc.? ratiometric normalization to ...?

More details have now been included in the methods (and below for the reviewer's convenience):

“For optimal image resolution, culture medium was replaced with Ham’s F-12 (Gibco) containing 25 mM HEPES for all live-cell imaging experiments. For BAY K8644 stimulation experiments, cells expressing the GFP-based calcium probe (pGP-CMV-GCaMP6s) were plated for at least 2 h on glass-bottom dishes (MatTek Corporation) precoated with 10 µg/ml of FN before the start of the treatment. Cells were then imaged every 10 s, on a TIRF microscope using either a 63x or 100x objective, at 37°C and using multi-position capabilities. BAY K8644 stimulation (1 µM final concentration) was performed live during imaging and maximal increases in calcium were observed after 1 min stimulation. Quantification of the changes in calcium concentration, based on the calcium probe intensity, was performed using ImageJ. GCaMP6s imaging: Laser line, 488 (2.8 % power); external filter, 490-550 nm; exposure time, 42 ms; EM gain, 101; large TIRF angles were used to ensure that only the cell-substrate interface was imaged. Definite focus was used to ensure that no variation occurred in the z position.

To study the relationship between filopodia stability and calcium concentration, cells were either co-expressing MYO10-mCherry and the GFP-based calcium probe (pGP-CMV-GCaMP6s) or expressing MYO10-mCherry and loaded with the calcium indicator dye Fluo-4 using the Fluo-4 Direct™ Calcium Assay Kit (ThermoFisher Scientific; F10471). When the GFP-based calcium probe was used cells were plated on FN for at least 2 h before the start of imaging. When Fluo-4 was used, cells were plated for 1 h on FN, before being loaded with the calcium dye according to the manufacturer’s instructions (30 min at 37°C + 30 min at room temperature). Cells were imaged within 1 h from the end of Fluo-4 loading as this dye rapidly disappeared from the TIRF plane and accumulates in intracellular bodies. For both the GFP-based calcium probe and the Fluo-4 approaches, cells were imaged live, every 5 s at 37°C, on a TIRF microscope using a 100x objective. GCaMP6s imaging: Laser line, 488 (5% power); external filter, 490-550 nm; exposure time, 85 ms; EM gain, 363; large TIRF angles were used to ensure that only the cell-substrate interface was imaged. Single position and definite focus were used to ensure that no variation occurred in the Z position. Fluo-4 imaging: Laser line, 488 (5% power); external filter, 490-550 nm; exposure time, 95 ms; EM gain, 66; large TIRF angles were used to ensure that only the cell-substrate interface was imaged. Single position and definite focus were used to ensure that no variation occurred in the Z position. The calcium probe / reporter intensity at stable (>1 min lifetime) and unstable filopodia (<1min lifetime) was measured using ImageJ. Briefly, for each cell, filopodia tips were detected, thresholded and their lifetime measured using the MYO10-mCherry channel. The integrated intensity of the calcium probe was measured at each filopodia tip. The measurements were then averaged in function of the filopodia lifetime and the ratio between calcium probe intensity at stable and unstable filopodia was calculated. The same

experiment and analysis was performed in cells expressing GFP and MYO10-mCherry as a negative control using the same settings as for the GCaMP6s imaging.”

Unfortunately, no attempt is made to calibrate calcium measurements. Why?

No calibration was performed as we are merely demonstrating relative increases in calcium concentration rather than attempting to prove accurate absolute values.

Does BayK induce an increase of Ca²⁺ in filopodia,

BayK stimulation triggers a transient increase in calcium in the cell body as well as in filopodia tips. To visualize the increase of calcium at filopodia tips, cells were imaged every 10 s, on a TIRF microscope using a 100x objective, at 37°C. BAY K8644 stimulation (1 μM final concentration) was performed live during imaging and maximal increases in calcium were observed after 1 min stimulation. GCaMP6s imaging: Laser line, 488 (2.8 % power); external filter, 490-550 nm; exposure time, 42 ms; EM gain, 101; large TIRF angles were used to ensure that only the cell-substrate interface was imaged. Definite focus was used to ensure that no variation occurred in the z position. These new data are presented in figure 1D and included below.

Figure 1D: MDA-MB-231 cells transiently expressing the calcium probe (pGP-CMV-GCaMP6s) and MYO10-mCherry were seeded on FN and treated with an L-type calcium channel activator (BAY K8644; 1 μM) while being imaged on a TIRF microscope (100x objective; scale bar = 10 μm). The inset shows a representative MYO10-positive filopodia tip delineated by a dotted line. ROI: region of interest.

and is prevented by the simultaneous application of amlodipine etc..

Simultaneous application of amlodipine besylate and BayK decreased the amplitude of the transient increase in calcium triggered by BayK. These new data are presented in Figure 1E (and included below).

Figure 1D: MDA-MB-231 cells transiently expressing the calcium probe (pGP-CMV-GCaMP6s) and adhering to FN were treated with an L-type calcium channel activator (BAY K8644; 1 μ M) in combination with DMSO or amlodipine besylate (1 μ M). The relative increase in the intracellular intensity of the calcium probe was measured at 1 min (three biological repeats, $n > 21$ cells; *** p value $< 1.39 \times 10^{-4}$).

Similarly, do the calcium channel blockers (in relevant concentrations) decrease Ca^{2+} in filopodia?

To further validate calcium channel blockers' ability to decrease calcium concentration at filopodia tips, we assessed the efficiency of 1 μ M amlodipine to inhibit calcium entry upon integrin activation. As shown in Supplementary Figure 8 (and the same data included below), 1 μ M amlodipine significantly reduces the signal intensity of the calcium probe at filopodia tips.

Supplementary Figure 8: MDA-MB-231 cells transiently expressing MYO10-mCherry and the calcium probe were plated on the anti-active $\beta 1$ integrin antibody (12G10) for 2 h in the presence of DMSO or 1 μ M amlodipine besylate. Representative images are displayed (scale bar = 20 μ m). For each conditions, the number of MYO10-positive filopodia per cell and the calcium probe intensity at filopodia tips were measured ($n > 85$ cells, three biological repeats; *** p value $< 3.2 \times 10^{-13}$).

Can this be mimicked by depolarising/hyperpolarising the cell membrane potential by changing the extracellular potassium concentration (in combination with potassium ionophores). Changing potassium is the classical approach to trigger L-type channel activity in smooth muscle research. Are these effects attenuated in siRNA treated cells?

To address this comment, we treated MDA-MB-231 cells, transiently expressing MYO10-mCherry, with 2.5 mM KCL or 40 mM KCL for 40 min. Increasing the extracellular potassium concentration did tend to increase the number of filopodia per cell, however overall this increase was very small, see below (figure included for reviewer only):

In presence of high extracellular potassium and Nigericin (potassium ionophore), Myo10-GFP starts to accumulate in the cells and forms large aggregates rendering the quantification of filopodia number impossible (see images below included for reviewer only). At the moment it is unclear why these aggregates are formed. One possible explanation is that treatment with the potassium ionophore (in presence of high extracellular potassium) triggers an equilibration between extracellular and intracellular pH, and we believe that MYO10 functions may be pH sensitive.

Nigericin

Altogether we find these two experiments unfortunately relatively inconclusive, so they were not added to the manuscript and are provided here as a response to the reviewer.

Omitting extracellular calcium is not specific because it also precludes calcium influx through a wide range of other calcium channels (TRPs, ORAI etc.) expressed in PDAC cells.

We agree with the reviewer that omitting extracellular calcium is not a specific way to inhibit L-type calcium channels. These experiments were performed only to assess the requirement of extracellular calcium in filopodia formation.

4. Antibody staining: How specific is the antibody? Given the notoriously poor quality of commercial antibodies against ion channels a control by Western blot etc. would have been nice.

The antibody used to stain CACNA1D appeared to be specific as the signal is very nicely decreased upon treatment with specific siRNA. These data are included in Supplementary Figure 6B and below. Unfortunately, we also found that this antibody does not work reliably in Western blots.

Supplementary Figure 6B: MDA-MB-231 cells previously silenced for CACNA1D were plated on FN and stained for CACNA1D (scale bar = 20 μ m). Average CACNA1D integrated density per cell was measured using ImageJ.

5. Who provides calcium for RAP activation; surprising that CCB's cannot be overcome by CA-Rap. Why does integrin activation increase calcium?

Calcium is only one of the ways by which the small GTPase Rap1 can be activated and Rap1 itself is not calcium sensitive (several Rap1 GEFs are calcium sensitive, e.g. CalDAG). Rap1 has also been

described to be activated by cAMP and several Rap1 GEFs are cAMP sensitive rather than calcium sensitive (e.g. Epac). Therefore it is possible that at filopodia Rap1 is activated via cAMP. In addition it is noteworthy that CalDAGs can be strongly activated by diacylglycerol. Further studies will be required to decipher the chemical signals that promote Rap1 activation in filopodia.

Our data demonstrate that integrin activation triggers an increase in calcium entry by activating L-type calcium channels (Figure 4) via downstream activation of Src (Figure 5). The pathway described here is most likely similar to the one described in smooth muscle cells (Work from MJ Davis laboratory, PMID:16554304 , PMID: 9763435, PMID: 11382763).

6. How can the invasion phenotype be distinguished from chemotaxis defects?

Chemotaxis is but one of the processes involved during invasion and therefore chemotaxis defects may result in invasion defects. However, work done by others suggests that filopodia only weakly contribute to chemotaxis but have a rather substantial impact on haptotaxis (PMID: 25666809). Our data suggest that CCB treatment inhibits directional cell migration on cell-derived matrices (Figure 2) and therefore would suggest that haptotaxis and possibly other forms of matrix sensing/grabbing/adhesion are inhibited here and are the primary contributors to filopodia-dependent invasion. However, we are not formally ruling out a possible contribution of chemotaxis as well.

Reviewer #2 (Remarks to the Author): Expert in cell motility

In this novel study, Jacquemet and colleagues use an inhibitor screen to identify a calcium-dependent pathway leading to filopodia formation and invasion of metastatic tumor cells. Further, they describe a detailed mechanism for how calcium entry through L-type calcium channels acts to stabilize focal adhesions downstream of integrin engagement. In the majority of the figures, the data is well presented and clear and supports the conclusions made by the author. In my opinion, this is an impressive body of work and it will be an important contribution to our understanding of how metastatic cells invade into the extracellular matrix. I only have a few questions and comments for the authors to address:

We thank the reviewer for his / her positive comments.

Questions and comments:

1. Clearly the Myo10-GFP is localizing to spots at the cell periphery. Are these spots always connected to the cell cortex via bundled F-actin as would be expected of filopodia? If so, why is this not seen in many of the images presented in the paper (eg. Figure 1A, 5B, S1E, S6A etc)?

Indeed the MYO10 spots are always connected to the cell cortex via actin. The reason why the actin is not always seen is that these structures are very small and often below the detection limits. This is especially true for the figures where cells were imaged using a 63x objective and the actin stained using a red dye (in these conditions, maximal Z resolution is around 211 nm, while filopodia are often between 50-150 nm). When cells expressing Myo10 are imaged using higher resolution (160X TIRF or STED) actin can be clearly observed in filopodia (see images below).

TIRF Microscope (160x)

STED Microscope

2. Does Myo10-GFP trigger increased filopodia formation in these cells? Are these induced filopodia functionally equivalent to the endogenous filopodia in terms of facilitating L-type channel-dependent invasion? An effective way to show they are functional would be to test if Myo10-GFP expression increased cell invasion dependent on L-type calcium channel activity.

Expression of MYO10 indeed promotes filopodia formation in all the cell lines tested. It is difficult to formally demonstrate that the MYO10-induced filopodia are fully equivalent to the endogenous filopodia or that even all filopodia are functionally equivalent. However, we know that silencing of endogenous MYO10 decreases the formation of endogenous filopodia and inhibits cell invasion in high-MYO10 expressing cancer cell lines (Arjonen et al., 2014 J. Clin. Invest.), suggesting that endogenous filopodia rely on MYO10. We have also shown that CCB treatment inhibits both the formation and the stability of MYO10-induced filopodia and endogenous filopodia (Figure 4F-G and Supplementary Figure 6D).

As suggested by the reviewer, we tested whether over-expression of MYO10 in a non-invasive cell line, would promote invasion. We found that over-expression of MYO10 in U2OS cells (U2OS MYO10-GFP cell line) promoted both filopodia formation and cancer cell invasion through collagen. Furthermore the ability of U2OS MYO10-GFP cells to invade was fully inhibited in the presence of amlodipine. These results are now included in the manuscript in Figure 2D and Supplementary Figures 5C-D and below for the reviewer's convenience.

Supplementary figure 5C and 5D: **C:** U2OS cells stably expressing GFP or MYO10-GFP were plated on FN and imaged live on a TIRF microscope. **D:** U2OS cells stably expressing GFP or MYO10-GFP were lysed and the levels of MYO10 were analysed by western blot. The uncropped blots are available in Supplementary Fig. 12.

Figure 2D: U2OS cells stably expressing either GFP or MYO10-GFP were seeded into an inverted invasion assay and allowed to invade for 4 days in the presence of amlodipine besylate (10 μM) or DMSO. Relative invasion over 45 μm was quantified (n = three biological repeats, *** p value < 4.05x10⁻⁶).

3. Pharmacological activation of the L-type calcium channels appears to increase calcium throughout the cell (Fig. 1C), but the voltage probe (Figure 1D) reports a voltage change specifically in filopodia. Can the authors account for this discrepancy and explain if the increase in calcium reported throughout the cell could be important for filopodia stabilization or invasive behavior?

Subcellular localization of the L-type calcium channel CACNA1D revealed that CACNA1D, localized not only to filopodia tips, but also at the plasma membrane (Figure 3D). Therefore it is not surprising that, upon treatment with an agonist (which opens L-type calcium channels regardless of the membrane potential), calcium entry will be observed throughout the cell including filopodia (see Figure 1D). We hypothesized that the voltage changes observed at the filopodia tips (using the voltage probe) could participate in the localized activation of these channels. We currently do not know whether activation of these channels, occurring in the cell body, contributes also to filopodia stabilization and / or cancer cell invasion.

4. What does time zero represent in Figure 1D? Is this prior to stimulation with the channel activator?

The time zero in Figure 1D (figure now removed) represented the beginning of the movie. No stimulation with BayK was performed in this experiment.

Again, does the increased signal in the cell body also contribute to filopodia stabilization? The right panel of figure 1D (tracing the changes in intensity over time) does not appear to show any clear differences between the cell body and filopodia and weakens the impact of the figure overall. The authors might consider removing it, or finding a way to more clearly portray the differences they measured.

Figure 1D (now removed) only aimed to indicate that variations in membrane potential at filopodia tips were observed. We thought that this observation was interesting and could be linked to localised calcium channel activation at filopodia tips. At this stage we do not know whether the variation in the cell body also influences filopodia stability. In addition we cannot experimentally prove that the variation in the membrane potential measured at filopodia tips is triggering filopodia stability. Therefore we decided to remove these data from the current manuscript.

5. FRET is not a trivial technique. Do the authors have the appropriate positive and negative controls to be able to confirm the decrease in the calpain FRET probe signal in filpodia is due to increased calpain activity (Fig. 6E)? This is an important experiment to get right, because it provides evidence for the specificity of action of calcium in the filopodia rather than in the cell body.

We agree with the reviewer that FRET is a difficult technic to set-up. As requested, we have provided additional controls showing that we can successfully detect FRET using the calpain probe. In particular, we compared the FRET signals in cells expressing the calpain probe or in cells expressing the donor or the acceptor alone or donor and acceptor together (free). To further validate that our technical set-up is capable of detecting biologically meaningful changes in calpain activation we have included several positive and negative controls. In particular, FRET signals were measured in cells previously silenced for calpain 1 and 2 and in cells treated with the calcium ionophore Calcimycin (A23187; calpain activator). We detected an increase in FRET signals when both calpain 1 and 2 were silenced and a decrease in FRET signals when calpains were activated using Calcimycin (see below). These data are presented in Supplementary Figure 11.

Supplementary Figure 11: A: MDA-MB-231 cells, previously silenced for calpain 1 and calpain 2, or treated with control oligo and transiently expressing CFP (donor only), YFP (acceptor only), CFP and YFP (free), or a calpain FRET probe (pCMV-calpainsensor; CFP and YFP linked by a calpain cleavage site, low FRET = higher calpain activity) were plated on FN and imaged on a confocal microscope. The averaged FRET signals measured in the cell body are displayed ($n > 34$; *** p value $< 8.9 \times 10^{-15}$). **B:** MDA-MB-231 cells transiently expressing a calpain FRET probe (pCMV-calpainsensor) and plated on FN were treated with DMSO or the calcium ionophore Calcimycin (to trigger calpain activation; A23187; 10 μ M) were imaged on a confocal microscope. The averaged FRET signals measured in the cell body are displayed ($n > 38$; *** p value $< 5.8 \times 10^{-11}$).

Reviewer #3 (Remarks to the Author): Expert in integrins and breast cancer

This paper from the Ivaska group presents data indicating that L-type calcium channels play a key role in the formation of filopodia and the establishment of invasive behaviour in cancer cells - in particular cancer cells whose invasiveness is driven by mutations in the p53 tumour suppressor. The authors show that four genes (CACNA1C, CACNA1D, CACNA1F & CACA1S) which encode the α 1-subunit of L-type Calcium channels are expressed in cancers and cancer cell lines, and that these channels are functional in a number of cancer cell lines which have mutated p53. They then proceed to show that calcium influx through these channels occurs in filopodia (particularly in the most stable filopodia) and that blockade, siRNA or CRISPR of channel subunits destabilises filopodia, and also opposes invasiveness. From this, the authors have performed a number of mechanistic analyses that establish a novel signalling pathway linking integrin activation to filopodial stability. The data support

the existence of a pathway that runs as follows:

Integrin engagement.....Rap1 activation.....integrin activation.....Src activation.....opening of L-type Ca channels.....calcium influx.....calpain activation.....filopodial stability

The experiments are generally well-executed and well-controlled. The findings will be of interest to the readership of Nat. Comms., and my feeling is that that paper can be published with little extra

experimentation. I do, however, have some reservations over the data that the authors use to justify a connection between expression of the CACNA1C , CACNA1D, CACNA1F & CACA1S genes and the aggressiveness of breast cancer. I think that this part of the paper needs more careful presentation of data and the statements concerning the strength of the clinical relevance of the expression of L-type calcium channels could be moderated.

We thank the reviewer for his / her positive comments.

I have the following comments which the authors might wish to consider:

1. The data presented in table 1 and in S3B puzzle me and I don't think that they necessarily support some of the authors' statements.

We agree with the reviewer and the conclusions based on these survival analyses have been tempered in the text (page 6).

A) Firstly, is it possible to take a number of genetic aberrations that affect channel subunits - e.g. mutation, deletion, amplification - and consider them as a single group entitled 'cases with alterations in L-type calcium channels'. Maybe I'm being naïve here, but surely deletion and amplification would be expected to have opposing affects.

We fully agree on this point. The grouping of these seemingly opposing genetic aberrations is linked to a technical limitation. The existing software to browse these publically available data did not allow us to distinguish between amplification and deletion. However, we concluded this to be a minor limitation as the number of deletions is very low in breast cancer samples (indicated in blue in Supplementary Figure 4A).

B) The difference between survival of patients without and with alteration to L-type Ca channels (as displayed in the KM plot in S3B and the 3rd from bottom row of table 1) appears (despite the low p value) to be very borderline. Is this really biologically significant? This is particularly in view of the

fact that the median survival time of cases with unaltered query genes (column 7 of table 1) is sometimes as low as 113 months which is so close to 101 months.

Indeed the differences observed in survival are not very big when all four L-type calcium channel subunits are analysed together. However, when the channels are analysed separately or in distinct combinations, the data suggests that CACNA1F expression does not appear to be associated with poor survival whereas CACNA1D is. The role of CACNA1C or CACNA1S remains unclear. These results are based on mRNA expression, data which might not reflect protein levels. To gain biological significance, it would be critical to further study the expression pattern of the various L-type calcium channel subunits in patient samples using immunohistochemistry in the future. We hope that the results provided in this paper will prompt further studies investigating the clinical relevance of the expression of these channels in other cancer types.

C) To me the result that is more exciting is the one in the 2nd row of Table 1 showing that 'altered' CACNA1D leads to substantially reduced survival times. What would a KM curve of this look like?

The KM curve for CACNA1D is now provided as Supplementary Figure 4D (and below).

Supplementary Figure 4D

D) Why has the median survival time of CACNA1F altered tumours not been determined and is displayed as NA?

This is the result given by the analysis software and is due to a very limited number of "altered case" deaths when CACNA1F is analysed.

D) Finally, in view of the above, I don't think that the authors can use the phrase 'Of the three clinically-relevant L-type calcium channel subunits' to justify their subsequent focus on 1D and 1S.

We have now removed this sentence.

2. The absolute values of the number of filopodia under control conditions seem to vary quite a lot in the paper. Sometimes this is about 10 per cell (Fig. 5A), and sometimes this is much higher (Figs, 1A, 2E). I suppose that these things do vary, but this causes problems when considering the effect of integrin-activating antibodies. The authors show that, when plated onto $\beta 1$ integrin activating antibodies, cells produce about 130-140 filopodia per cell. This is fine, but the (carefully named!) anti-inactive $\beta 1$ antibody leads to at least 60 filopodia per cells, which is many more than observed in control cells.

These numbers cannot be directly compared as the cells have been imaged and quantified differently. The “control condition” mentioned here corresponds to cells plated on FN for 2 hours and imaged using a 63x objective and the number of filopodia per cell quantified manually. When plated on anti-integrin antibodies (anti-active or anti-inactive) the cells were imaged using a 100x objective and the number of filopodia per cell was quantified automatically using an in-house software (provided as a supplementary file). These two approaches yield different number of filopodia per cell, mostly due to the fact that using the 100x objective more filopodia can be more clearly observed and detected. When compared directly (using 100x objective and automated quantification), cells plated on FN actually yield a similar number of filopodia compared to cells plated on the anti-inactive integrin antibody (see below as additional data for reviewer#3). This figure has not been included in the manuscript but could be added if considered essential.

We believe that cells plated on the anti-inactive integrin antibody display many filopodia as our unpublished observations (see below) suggest that filopodia are decorated by inactive integrin and therefore these filopodia are captured by the antibody coating. This notion is supported by the fact that filopodia are completely immobilized in cells plated on the anti-inactive integrin antibody, according to live-cell imaging of MYO10-GFP. Displayed here, for the reviewer, are immunofluorescence staining of inactive integrin in filopodia (see below):

Figure: STED imaging of inactive $\beta 1$ integrin in a cell overexpressing MYO10-GFP and plated on FN.

The authors perhaps need to directly compare no antibody (or irrelevant antibody), anti-inactive $\beta 1$ and anti-active $\beta 1$ directly to illustrate the point that immobilisation of integrins with either of the these antibodies tends to increase filopodia number, but to different extents.

To answer this question we compared the number of MYO10-positive filopodia in cells plated on anti-integrin antibodies (anti-active and anti-inactive) to cells plated on Poly-L-lysine or anti-transferrin receptor antibodies for 2h. In comparison to these two conditions (PolyL or anti-transferrin receptor antibodies), cells plated on anti-inactive $\beta 1$ integrin antibody yielded more filopodia (see below for reviewer only). But this is due to the fact that PolyL and anti-transferrin

receptor antibodies failed to immobilise filopodia (as well as promote cell spreading). Altogether the important conclusion is that integrin activation (i.e. signalling) rather than immobilisation is able to trigger increases in filopodia formation. This figure for reviewer only has not been included in the manuscript but could be added if requested.

3. Fig 1B needs some expanded panels so that expression of the channel subunits in the named cell lines can be seen.

The figure has now been extended so the expression pattern of the L-type calcium channel subunits in the annotated cell lines are visible (new Figure 1B and shown below).

Figure 1B

4. The paper is left a little hanging at the end and the authors' assertion that calpain activation by calcium entry through L-type channels leads to maturation of filopodia into FAs has not been

developed. Questions that spring to mind are: What is the effect of blockade of L-type channels on FA formation as cell migrate?

To answer this question, cells transiently expressing paxillin-GFP were plated on FN, treated with DMSO or amlodipine and left to migrate while being imaged on a TIRF microscope. Focal adhesion dynamics including kinetics of focal adhesion formation and disassembly were then analysed using the focal adhesion analysis server. CCB treatment only weakly impacted focal adhesion dynamics as their overall lifetime distribution was unaffected. CCB treatment appeared to decrease both the assembly and disassembly rates of focal adhesions as well as their maximal size. These data have now been included in Figure 8A (and below).

Figure 8A. MDA-MB-231 cells transiently expressing mEmerald-Paxillin were plated on FN and imaged live using a TIRF microscope (1 picture every 1 min over 3 h; scale bar = 20 μm) in the presence of DMSO or amlodipine besylate (10 μM). Focal adhesion properties were analysed using the focal adhesion analysis server (three biological repeats; over 33 movies per condition analysed; adhesion lifetime and maximal area, n > 112000 adhesions analysed; assembly rate, n > 6100 adhesions analysed; disassembly rate, n > 7200 adhesions analysed, *** p value < 1.73x10⁻⁵⁰).

What is the role of calpain-mediated talin cleavage in this? Some of this might be outwith the scope of the paper, but the first question should be relatively easy to address.

We have been trying hard to identify the calpain substrate(s) involved in promoting filopodia formation. We do not believe that talin is directly involved as over-expression of talin head (fragment cleaved upon calpain-2 proteolysis which also drives integrin activation and induces filopodia) does not rescue the filopodia formation downstream of CCB or calpain inhibitor treatment (see below).

We also looked into other known calpain targets including paxillin, FAK and the phosphatase PTP1B but none of these appear to be involved either.

5. There are no scale bars on the IHC

We thank the reviewer for noticing this. We have now provided a scale bar for these images.

6. What are the very stable spots in movie 5? (FEL-treated). They don't look like filopodia and their presence rather undercuts the authors' claims that FEL opposes FP stability.

These very stable spots appeared to be cell aggregates. Even though these spots are not filopodia, they would have been detected by the automated tracking software used to quantify filopodia dynamics and do not change our conclusions. However, we agree with the reviewer that this movie is maybe not the best one to display and therefore we have now provided a new one and Figure 4C has been modified accordingly (see below).

Figure 4C.

7. Fig 5C doesn't look particularly convincing as displayed. It's not really possible to see the Myo10 and active colocalisation.

This figure (now Figure 6C) aimed to indicate that active Src localizes to filopodia tips and filopodia shafts rather than suggesting that active Src co-localises with MYO10. The text has now been modified to clearly reflect this.

8. '60% less filopodia' should read '60% fewer filopodia'

This has now been corrected.

REVIEWERS' COMMENTS:

Reviewer #1 (Remarks to the Author):

All concerns have been addressed adequately. I have no further comments.

Reviewer #2 (Remarks to the Author):

The authors have addressed all of my concerns and I recommend acceptance of their manuscript. This paper will make an important contribution to our understanding of how protrusions are generated in metastatic cells to drive their invasive behavior. I look forward to the publication of this ground-breaking work.

Reviewer #3 (Remarks to the Author):

I have reviewed the revisions and rebuttal that the authors have produced in response to my original comments. As I stated previously, I considered that this paper was largely acceptable for publication. In view of the authors' responses and the modifications that they have made, that the paper is now acceptable for publication in Nat. Comms.

I would like to commend the authors on a very interesting and convincing study, highlighting a novel mechanism through which breast cancer invasiveness is controlled.

REVIEWERS' COMMENTS:

Reviewer #1 (Remarks to the Author):

All concerns have been addressed adequately. I have no further comments.

We thank the reviewer for assessing our manuscript.

Reviewer #2 (Remarks to the Author):

The authors have addressed all of my concerns and I recommend acceptance of their manuscript. This paper will make an important contribution to our understanding of how protrusions are generated in metastatic cells to drive their invasive behavior. I look forward to the publication of this ground-breaking work.

We thank the reviewer for his / her positive comments.

Reviewer #3 (Remarks to the Author):

I have reviewed the revisions and rebuttal that the authors have produced in response to my original comments. As I stated previously, I considered that this paper was largely acceptable for publication. In view of the authors' responses and the modifications that they have made, that the paper is now acceptable for publication in Nat. Comms.

I would like to commend the authors on a very interesting and convincing study, highlighting a novel mechanism through which breast cancer invasiveness is controlled.

We thank the reviewer for his / her positive comments.